# Convolutional Deep Kernel Machines

**Edward Milsom**
School of Mathematics
University of Bristol
`edward.milsom@bristol.ac.uk`

**Ben Anson**
School of Mathematics
University of Bristol
`ben.anson@bristol.ac.uk`

**Laurence Aitchison**
Department of Computer Science
University of Bristol
`laurence.aitchison@gmail.com`

## Abstract

Standard infinite-width limits of neural networks sacrifice the ability for intermediate layers to learn representations from data. Recent work ("A theory of representation learning gives a deep generalisation of kernel methods", Yang et al. 2023) modified the Neural Network Gaussian Process (NNGP) limit of Bayesian neural networks so that representation learning is retained. Furthermore, they found that applying this modified limit to a deep Gaussian process gives a practical learning algorithm which they dubbed the "deep kernel machine" (DKM). However, they only considered the simplest possible setting: regression in small, fully connected networks with e.g. 10 input features. Here, we introduce convolutional deep kernel machines. This required us to develop a novel inter-domain inducing point approximation, as well as introducing and experimentally assessing a number of techniques not previously seen in DKMs, including analogues to batch normalisation, different likelihoods, and different types of top-layer. The resulting model trains in roughly 77 GPU hours, achieving around 99% test accuracy on MNIST, 72% on CIFAR-100, and 92.7% on CIFAR-10, which is SOTA for kernel methods.

## 1 Introduction

A key theoretical approach to studying neural networks is taking the infinite-width limit of a randomly initialised network. In this limit, the outputs become Gaussian process (GP) distributed, and the resulting GP is known as an NNGP (Neal, 1995; Lee et al., 2017; Matthews et al., 2018). However, despite a large body of work attempting to improve the predictive performance of NNGPs (Novak et al., 2018; Garriga-Alonso et al., 2018; Arora et al., 2019; Lee et al., 2020; Li et al., 2019; Shankar et al., 2020; Adlam et al., 2023), they still empirically underperform finite NNs. Understanding why this gap exists will improve our understanding of neural networks. One hypothesis (Aitchison, 2020; MacKay, 1998) is that the NNGP lacks representation learning, as the NNGP kernel is a fixed and deterministic function of the inputs, while the top-layer representation in a finite neural network is learned; representation learning is believed to be critical to the success of modern deep learning (Bengio et al., 2013; LeCun et al., 2015). One way to confirm this hypothesis would be to find an NNGP-like approach that somehow incorporates representation learning, and gives improved performance over the plain NNGP. While some theoretical approaches to representation learning have been developed (Antognini, 2019; Dyer & Gur-Ari, 2019; Hanin & Nica, 2019; Aitchison, 2020; Li & Sompolinsky, 2020; Yaida, 2020; Naveh et al., 2020; Zavatone-Veth et al., 2021; Zavatone-Veth & Pehlevan, 2021; Roberts et al., 2021; Naveh & Ringel, 2021; Halverson et al., 2021; Seroussi et al., 2023), they are not scalable enough to apply to many common datasets (e.g. Seroussi et al., 2023 could only consider a subset of 2048 points from CIFAR-10). Here, we develop the convolutional deep kernel machine (DKM), an NNGP-like theoretical approach which captures representation learning and can be scaled to datasets like CIFAR-10. Convolutional DKMs reduce (but still do not fully close) the gap between theoretical approaches and real neural networks, giving further evidence that representation learning is the critical component missing from approaches like the NNGP.

Our theoretical approach is based on DKMs (Yang et al., 2023). DKMs are fundamentally a *deep* Gaussian process (DGP), where the infinite-width limit has been taken carefully, to ensure that representation learning is retained. While a DGP is not a NN, they are very similar: both DGPs and NNs are deep, nonlinear function approximators that admit representation learning, and many formal connections between NNs and DGPs have been drawn in the literature (Dutordoir et al., 2021; Agrawal et al., 2020; Pleiss & Cunningham, 2021). Furthermore, DGP-based theoretical approaches (i.e. DKMs) offer considerable simplifications over NN-based approaches. These mostly arise from the fact that the posterior over features in an infinite-width DGP is exactly multivariate Gaussian, even when that infinite-width limit retains representation learning. This is an important factor in the scalability of DKMs.

Additionally, Yang et al. (2023) developed inducing-point methods for DKMs inspired by the DGP literature, making them particularly scalable. Naive, full-rank DKMs have cubic time-complexity in the number of training points, which is intractable in all but the smallest datasets. By contrast, inducing-point DKMs require linear compute in the number of datapoints (and cubic compute in the far smaller number of inducing points). However, Yang et al. (2023) only considered regression in small fully-connected architectures with e.g. 10 input features. To compare the performance of DKMs to CNNs on datasets such as CIFAR-10, we must develop convolutional DKMs.

However, developing an efficient, scalable DKM inducing point scheme for the convolutional setting is highly non-trivial. Naively applying the inducing point inference scheme from Yang et al. (2023) implies taking images as inducing points, resulting in prohibitive $\mathcal{O}(P_i^3 W^3 H^3)$ operations, where $P_i$ is the number of inducing points, and $W, H$ are the width and height of the images. One alternative is inducing patch schemes from the GP literature (van der Wilk et al., 2017; Blomqvist et al., 2018; Dutordoir et al., 2020); however they cannot be applied to DKMs, as they rely on intermediate features, whereas DKMs only propagate kernel/Gram matrices through the layers, *not* features. We were thus forced to develop an entirely new inter-domain (Lázaro-Gredilla & Figueiras-Vidal, 2009; Hensman et al., 2017; Rudner et al., 2020) convolutional inducing point scheme, designed specifically for the DKM setting. To summarise, our contributions are:

- Introducing convolutional DKMs, which achieve SOTA performance for kernel methods on CIFAR-10 with an accuracy of 92.7%.
- Introducing an efficient inter-domain inducing point scheme for convolutional DKMs, which allows our largest model to train in 77 GPU hours, 1–2 orders of magnitude faster than full NNGP / NTK / Myrtle kernels.
- Developing a number of different model variants, empirically investigating their performance, including various normalisation schemes (Appendix C), two likelihoods (Appendix D.2) and two different top-layers (Appendix F.3).

## 2   RELATED WORK

DKMs were developed in  Yang et al. (2023), using a modified NNGP limit that retains representation learning. However, they only considered regression in small fully-connected networks.

The NTK setting (Jacot et al., 2018) is a theoretical approach to understanding neural networks that is also based on infinite-width limits. However, the NTK deals with very different phenomena from the NNGP/DKM setting, in that it describes the dynamics of gradient descent applied to a loss function. By contrast, the NNGP/DKM (Yang et al., 2023) describes Bayesian networks under the posterior, which does not, in principle, involve any dynamics. A problem arises in the NTK setting, whereby the NTK, describing the dynamics of gradient descent, is constant through training in particular infinite-width limits. Recent work on the mu-P scaling has resolved this issue, by finding alternative limits in which the NTK is able to evolve through training (Yang & Hu, 2021; Yang et al., 2022).

There is a body of literature on infinite-width convolutional neural networks (CNNs) (Novak et al., 2018; Garriga-Alonso et al., 2018), specifically in the NNGP setting (Neal, 1995; Lee et al., 2017; Matthews et al., 2018). As mentioned before, NNGPs have a kernel that is a fixed, deterministic function of the inputs, and therefore do not allow for flexibility in the learned kernels/representations, unlike real, finite CNNs and unlike our convolutional DKMs (Aitchison, 2020; Yang et al., 2023).

Convolutional DGPs use inducing patches for scalability (van der Wilk et al., 2017; Blomqvist et al., 2018; Dutordoir et al., 2020). However, we cannot use their scheme here, as it requires features

at each layer, while we work solely with Gram matrices. Therefore we were forced to develop a new scheme. Our DKM inducing point scheme is inter-domain (Lázaro-Gredilla & Figueiras-Vidal, 2009; Hensman et al., 2017; Rudner et al., 2020), in the sense that the inducing points do not mirror datapoints (images), but instead store information about the function in a different domain. Existing work on inter-domain inducing points does not address the DKM or convolutional settings.

An alternative approach to getting a flexible kernel is to take the inputs, transform them through a NN (e.g. 10–40 layer CNN), then to use the outputs of the NN as inputs to a standard kernel. This approach is known as deep kernel learning (DKL; Calandra et al., 2016; Wilson et al., 2016b;a; Bradshaw et al., 2017; Bohn et al., 2019; Ober et al., 2021). DKMs are very different to DKL in that there is no underlying neural network. Instead the DKM directly transforms and allows flexibility in the kernels.

There are a small number of other "deep kernel" approaches such as convolutional kernel machines, (Mairal et al., 2014; Mairal, 2016; Suykens, 2017; Chen et al., 2019; Tonin et al., 2023; Achten et al., 2023). These differ from our approach in that they lack the theoretical connection to representation learning in NNGPs, and thus cannot answer the questions raised in the Introduction. Moreover, many of these methods use a neural-network like architecture to give a flexible, feature-based approximation to an underlying fixed kernel, whereas our approach fundamentally allows for flexibility in the underlying kernel/representation; we believe this explains our performance advantage (Table 3). Nonetheless, there are fruitful connections to be drawn between NNGPs, physics-based theory (e.g. Seroussi et al., 2023) and e.g. convolutional kernel networks (Mairal et al., 2014; Mairal, 2016), and we hope that our work contributes to fostering a dialogue between these areas.

## 3 BACKGROUND

**Deep Kernel Machines.** A detailed derivation of DKMs can be found in Appendix A or Yang et al. (2023). In a nutshell, DKMs are derived from deep Gaussian processes where the intermediate layers $\mathbf{F}^\ell \in \mathbb{R}^{P \times N_\ell}$, with $P$ datapoints and $N_\ell$ features, have been made infinitely wide ($N_\ell \to \infty$). However, in the traditional infinite-width limit, representations become fixed (e.g. Aitchison, 2020; Yang et al., 2023). The DKM (Yang et al., 2023) modifies the likelihood function so that, in the infinite limit, representations remain flexible and adapt to data. While the full derivation is somewhat involved, the key idea of the resulting algorithm is to consider the Gram matrices (which are $P \times P$ matrices like kernels) at each layer of the model as *parameters*, and optimise them according to an objective. Here we give a short, practical introduction to DKMs.

As DKMs are ultimately infinite-width DGPs, and we cannot practically work with infinite-width features, we must instead work in terms of Gram matrices:

$$\mathbf{G}^\ell = \tfrac{1}{N_\ell} \mathbf{F}^\ell (\mathbf{F}^\ell)^T \in \mathbb{R}^{P \times P}. \tag{1}$$

Note that from this point on we will always refer to $\mathbf{G}^\ell$ in Eq. (1) as a "Gram matrix", whilst the term "kernel matrix" will carry the usual meaning of a matrix of similarities computed using a kernel function. Ordinarily, at each layer, a DGP computes the kernel matrix $\mathbf{K}_{\text{features}}(\mathbf{F}^{\ell-1}) \in \mathbb{R}^{P \times P}$ from the previous layer's features. Since we no longer have access to $\mathbf{F}^{\ell-1}$, but instead propagate $\mathbf{G}^{\ell-1}$, computing $\mathbf{K}_{\text{features}}(\mathbf{F}^{\ell-1})$ is not possible in general. However, it turns out that for many kernels of interest, we can compute $\mathbf{K}$ from $\mathbf{G}^{\ell-1}$. For example, isotropic kernels (such as the squared exponential) only depend on $\mathbf{F}$ through the normalised squared distance $R_{ij}$ between points, which can be computed from the Gram matrices,

$$R_{ij} = \tfrac{1}{N_\ell} \textstyle\sum_{\lambda=1}^{N_\ell} (F_{i\lambda} - F_{j\lambda})^2 = \tfrac{1}{N_\ell} \textstyle\sum_{\lambda=1}^{N_\ell} \left( F_{i\lambda}^2 - 2 F_{i\lambda} F_{j\lambda} + F_{j\lambda}^2 \right) = G_{ii}^\ell - 2 G_{ij}^\ell + G_{jj}^\ell, \quad (2)$$

and hence we can instead compute the kernel matrix as

$$\mathbf{K}(\mathbf{G}^{\ell-1}) = \mathbf{K}_{\text{features}}(\mathbf{F}^{\ell-1}). \tag{3}$$

Here, $\mathbf{K}(\cdot)$ and $\mathbf{K}_{\text{features}}(\cdot)$ are functions that compute the same kernel, but $\mathbf{K}(\cdot)$ takes a Gram matrix as input, whereas $\mathbf{K}_{\text{features}}(\cdot)$ takes features as input. Notice that $\mathbf{K}(\mathbf{G}^{\ell-1}) \in \mathbb{R}^{P \times P}$ has the same shape as $\mathbf{G}^{\ell-1}$, and so it is possible to recursively apply the kernel function in this parametrisation, taking $\mathbf{G}^0 = \tfrac{1}{N_0} \mathbf{X} \mathbf{X}^T$ where $\mathbf{X} \in \mathbb{R}^{P \times N_0}$ is the input data:

$$\mathbf{G}^\ell = \mathbf{K}(\mathbf{G}^{\ell-1}) = \underbrace{\mathbf{K}(\mathbf{K}(\cdots \mathbf{K}(\mathbf{G}^0)))}_{\ell \text{ times}}. \tag{4}$$

It turns out Eq. (4) exactly describes the traditional infinite-width limit of a DGP (i.e. the NNGP limit). Notice that $\mathbf{G}^\ell$ is a fixed function of $\mathbf{G}^0$ and thus cannot adapt to data (perhaps with the exception of a small number of tunable kernel hyperparameters). This is a major disadvantage compared to finite NNs and DGPs, which flexibly learn representations at each layer from the data.

The DKM solves this problem by taking a modified limit, the "Bayesian representation learning limit" (Yang et al., 2023). Under this alternative limit, the Gram matrices have some flexibility, and are no longer fixed and equal to those given by Eq. (4). Instead, the $L$ Gram matrices, $\mathbf{G}^1, \ldots, \mathbf{G}^L$, become parameters which are optimised by maximising the DKM objective,

$$\mathcal{L}(\mathbf{G}^1, \ldots, \mathbf{G}^L) = \log \mathrm{P}\left(\mathbf{Y}|\mathbf{G}^L\right) - \sum_{\ell=1}^L \nu_\ell \, \mathrm{D}_{\mathrm{KL}}\left(\mathcal{N}\left(\mathbf{0}, \mathbf{G}^\ell\right) \| \mathcal{N}\left(\mathbf{0}, \mathbf{K}(\mathbf{G}^{\ell-1})\right)\right). \quad (5)$$

The first term encourages good predictions by maximising the log-likelihood of the training data $\mathbf{Y}$, whilst the other KL-divergence terms form a regulariser that encourages $\mathbf{G}^\ell$ to be close to $\mathbf{K}(\mathbf{G}^{\ell-1})$. The model reduces to the standard infinite-width limit (Eq. 4) when no data is observed, since the KL-divergence is minimised by setting $\mathbf{G}^\ell = \mathbf{K}(\mathbf{G}^{\ell-1})$. Formally, the DKM objective is the evidence lower bound (ELBO) for an infinite-width DGP in the Bayesian representation learning limit. After optimising the training Gram matrices, prediction of unseen test points (which requires us to predict the test-test and test-train Gram matrix blocks for each layer progressively before predicting the final layer output) proceeds using an algorithm inspired by DGPs (see Yang et al. 2023 for details).

**Sparse DKMs.** As with all naive kernel methods, directly computing the DKM objective for all data is $\mathcal{O}(P_\mathrm{t}^3)$, where $P_\mathrm{t}$ is the number of datapoints, which is typically infeasible. As a DKM is ultimately an infinite-width DGP parameterised by Gram matrices rather than features, Yang et al. (2023) developed an inducing point scheme inspired by the DGP literature.

In the GP context, inducing point schemes reduce computational complexity by using variational inference with an approximate posterior that replaces the training set (of size $P_\mathrm{t}$) with a set of $P_\mathrm{i}$ pseudo-inputs, called inducing points, where $P_\mathrm{i}$ is usually much smaller than $P_\mathrm{t}$. The inducing points can be learned as variational parameters. In DGPs, each layer is a GP, so we learn one set of inducing points for each layer. When using inducing points, the features at each layer are partitioned into inducing points $\mathbf{F}_\mathrm{i}^\ell$ and training points $\mathbf{F}_\mathrm{t}^\ell$. Likewise, the kernel matrix is partitioned into blocks corresponding to inducing and training points. In a DKM, we partition $\mathbf{G}^\ell$ similarly,

$$\mathbf{G}^\ell = \begin{pmatrix} \mathbf{G}_\mathrm{ii}^\ell & \mathbf{G}_\mathrm{it}^\ell \\ \mathbf{G}_\mathrm{ti}^\ell & \mathbf{G}_\mathrm{tt}^\ell \end{pmatrix} \qquad \mathbf{K}(\mathbf{G}^\ell) = \begin{pmatrix} \mathbf{K}_\mathrm{ii}(\mathbf{G}_\mathrm{ii}^\ell) & \mathbf{K}_\mathrm{it}(\mathbf{G}^\ell) \\ \mathbf{K}_\mathrm{ti}(\mathbf{G}^\ell) & \mathbf{K}_\mathrm{tt}(\mathbf{G}_\mathrm{tt}^\ell) \end{pmatrix}, \quad (6)$$

with $\mathbf{G}_\mathrm{ii}^\ell \in \mathbb{R}^{P_\mathrm{i} \times P_\mathrm{i}}$, $\mathbf{G}_\mathrm{ti}^\ell \in \mathbb{R}^{P_\mathrm{t} \times P_\mathrm{i}}$, and $\mathbf{G}_\mathrm{tt}^\ell \in \mathbb{R}^{P_\mathrm{t} \times P_\mathrm{t}}$. At each layer $\ell$, $\mathbf{G}_\mathrm{ii}^\ell$ is learnt, whilst $\mathbf{G}_\mathrm{ti}^\ell$ and $\mathbf{G}_\mathrm{tt}^\ell$ are predicted using $\mathbf{G}_\mathrm{ii}^\ell$ and $\mathbf{K}(\mathbf{G}^{\ell-1})$ (see Yang et al., 2023). The sparse DKM objective is

$$\mathcal{L}_\mathrm{ind}(\mathbf{G}_\mathrm{ii}^1, \ldots, \mathbf{G}_\mathrm{ii}^L) = \text{GP-ELBO}\left(\mathbf{Y}_\mathrm{t}; \mathbf{G}_\mathrm{ii}^L, \mathbf{G}_\mathrm{ti}^L, \mathbf{G}_\mathrm{tt}^L\right) - \sum_{\ell=1}^L \nu_\ell \, \mathrm{D}_{\mathrm{KL}}\left(\mathcal{N}\left(\mathbf{0}, \mathbf{G}_\mathrm{ii}^\ell\right) \| \mathcal{N}\left(\mathbf{0}, \mathbf{K}(\mathbf{G}_\mathrm{ii}^{\ell-1})\right)\right).$$
$$(7)$$

where GP-ELBO $\left(\mathbf{Y}_\mathrm{t}; \mathbf{G}_\mathrm{ii}^L, \mathbf{G}_\mathrm{ti}^L, \mathbf{G}_\mathrm{tt}^L\right)$ is the ELBO for a sparse GP with data $\mathbf{Y}_\mathrm{t}$ and kernel $\mathbf{K}(\mathbf{G}_L)$, and contains a likelihood term which measures the performance of the model on training data. We consider two different likelihoods: categorical and Gaussian (see Appendix D for further details). This scheme is efficient, with time complexity $\mathcal{O}(L(P_\mathrm{i}^3 + P_\mathrm{i}^2 P_\mathrm{t}))$, both because the KL terms in the sparse DKM objective depend only on $\mathbf{G}_\mathrm{ii}$, and because obtaining predictions and computing the performance term in the DKM objective requires only the diagonal of $\mathbf{G}_\mathrm{tt}^\ell$ (which mirrors the requirements for the GP ELBO; see Yang et al., 2023 for details).

## 4 METHODS

The overall approach to defining a convolutional DKM is to build convolutional structure into our kernel, $\mathbf{K}(\cdot)$, by taking inspiration from the infinite-width convolutional neural network literature (Novak et al., 2018; Garriga-Alonso et al., 2018). Specifically, we will define the kernel to be the covariance between features induced by a single convolutional BNN layer.

Take a convolutional BNN layer, with inputs $\mathbf{H}^{\ell-1} \in \mathbb{R}^{PS_{\ell-1} \times N_{\ell-1}}$ and outputs $\mathbf{F}^\ell \in \mathbb{R}^{PS_\ell \times N_\ell}$. Here, $P$ is the number of images, $N_\ell$ is the number of output channels at layer $\ell$, and $S_\ell$ is the number of output spatial locations in the feature map (e.g. $S_\ell = W_\ell \times H_\ell$ for a 2D image). The convolutional

weights are $\mathbf{W}^\ell \in \mathbb{R}^{D \times N_{\ell-1} \times N_\ell}$, and $D$ is the number of spatial locations in a filter (e.g. a $3 \times 3$ convolutional patch would have $D = 9$). The BNN convolution operation (i.e. `Conv2d` in PyTorch) can be written as

$$F^\ell_{ir,\lambda} = \sum_{d \in \mathcal{D}} \sum_{\mu=1}^{N_{\ell-1}} H^{\ell-1}_{i(r+d),\mu} W^\ell_{d\mu\lambda}. \tag{8}$$

Here, $i$ indexes the image, $r$ the spatial location in the image, $d$ the location in the patch, and $\mu$ and $\lambda$ the input and output channels, respectively. This is natural for 1D convolutions, e.g. $r \in \{1, \ldots, S_\ell\}$ and $d \in \mathcal{D} = \{-1, 0, 1\}$ (with $D = 3$ in this example), but also generalises to higher dimensions, e.g. using vectors. In a typical CNN, we would apply a nonlinearity $\phi$, e.g. relu, to the output of the previous layer to obtain the input to the next layer, i.e. $\mathbf{H}^{\ell-1} = \phi(\mathbf{F}^{\ell-1})$.

We will define the kernel as the covariance between the outputs of the convolutional layer (Eq. 8), $\Gamma_{ir,js} = \mathbb{E}\left[F^\ell_{ir,\lambda} F^\ell_{js,\lambda} | \mathbf{H}^{\ell-1}\right]$, where we have placed an IID Gaussian prior on the weights:

$$\mathrm{P}\left(W^\ell_{d\mu\lambda}\right) = \mathcal{N}\left(W^\ell_{d\mu\lambda}; 0, \tfrac{1}{DN_{\ell-1}}\right). \tag{9}$$

Note that this implies all features (indexed by $\lambda$) are IID, so we consider an arbitrary feature $\lambda$. This is valid even if $N_\ell \to \infty$. Since the weights have zero mean, the distribution over features (Eq. 8) has zero mean, so $\Gamma_{ir,js}$ is just the conditional covariance of the output features. To compute this covariance it will be useful to define a Gram-like matrix $\boldsymbol{\Omega}^{\ell-1} \in \mathbb{R}^{PS_{\ell-1} \times PS_{\ell-1}}$ from the input $\mathbf{H}^\ell$,

$$\boldsymbol{\Omega}^{\ell-1} = \tfrac{1}{N_{\ell-1}} \mathbf{H}^{\ell-1} \left(\mathbf{H}^{\ell-1}\right)^T \in \mathbb{R}^{PS_{\ell-1} \times PS_{\ell-1}}, \qquad \Omega^{\ell-1}_{ir,js} = \tfrac{1}{N_{\ell-1}} \sum_{\mu=1}^{N_{\ell-1}} H^{\ell-1}_{ir,\mu} H^{\ell-1}_{js,\mu} \tag{10}$$

For the matrix multiplication to make sense, we have interpreted $\mathbf{H}_{\ell-1} \in \mathbb{R}^{PS_{\ell-1} \times N_{\ell-1}}$ as a matrix with $PS_{\ell-1}$ rows and $N_{\ell-1}$ columns. The covariance $\Gamma_{ir,js}$ can be computed (Appendix B.3) as,

$$\Gamma_{ir,js}(\boldsymbol{\Omega}^{\ell-1}) = \mathbb{E}\left[F^\ell_{ir,\lambda} F^\ell_{js,\lambda} | \mathbf{H}^{\ell-1}\right] = \tfrac{1}{D} \sum_{d \in \mathcal{D}} \Omega^{\ell-1}_{i(r+d),j(s+d)}. \tag{11}$$

This is not yet a complete kernel function for our convolutional DKM since it does not take the previous layer Gram matrix $\mathbf{G}^{\ell-1}$ (Eq. 1) as input. To complete the kernel, recall that $\mathbf{H}^{\ell-1}$ is defined by applying e.g. a ReLU nonlinearity to $\mathbf{F}^{\ell-1}$. Taking inspiration from infinite NNs, that would suggest we can compute $\boldsymbol{\Omega}^{\ell-1}$ from $\mathbf{G}^{\ell-1}$ using an arccos kernel (Cho & Saul, 2009),

$$\boldsymbol{\Omega}^{\ell-1} = \boldsymbol{\Omega}(\mathbf{G}^{\ell-1}). \tag{12}$$

Here, $\boldsymbol{\Omega}(\cdot)$ is a function that applies e.g. an arccos kernel to a Gram matrix, while $\boldsymbol{\Omega}^{\ell-1}$ is the specific value at this layer. Combining this with Eq. (11), we obtain the full convolutional DKM kernel,

$$\mathbf{K}_{\mathrm{Conv}}(\mathbf{G}^{\ell-1}) = \mathbb{E}\left[\mathbf{f}^\ell_\lambda \left(\mathbf{f}^\ell_\lambda\right)^T | \mathbf{H}^{\ell-1}\right] = \boldsymbol{\Gamma}(\boldsymbol{\Omega}(\mathbf{G}^{\ell-1})) \in \mathbb{R}^{PS_\ell \times PS_\ell}, \tag{13}$$

where in Eq. (13) we have written the equation for the entire matrix instead of a specific element as in Eq. (11). Here, $\mathbf{f}^\ell_\lambda \in \mathbb{R}^{PS_\ell}$ denotes a single feature vector, i.e. the $\lambda^{\text{th}}$ column of $\mathbf{F}^\ell$. From this perspective, $\boldsymbol{\Gamma}(\cdot)$ (Eq. 11) can be viewed as the kernelised version of the convolution operation.

**Sparse convolutional DKMs.** Computing the full kernel for all training images is intractable in the full-rank case. For example, CIFAR-10 has $50\,000$ training images of size $32 \times 32$, so we would need to invert a kernel matrix with $50\,000 \cdot 32 \cdot 32 = 51\,200\,000$ rows and columns. Hence, we need a more efficient scheme. Following Yang et al. (2023), we consider a sparse inducing point DKM. However, inducing points usually live in the same space as datapoints. If we take inducing points as images, we end up with $\mathbf{G}^\ell_{\mathrm{ii}} \in \mathbb{R}^{S_\ell P_{\mathrm{i}} \times S_\ell P_{\mathrm{i}}}$. This is still intractable, even for a small number of images. For CIFAR-10, $S = 32 \cdot 32 = 1024$, so with as few as $P_{\mathrm{i}} = 10$ inducing points, $\mathbf{G}^\ell_{\mathrm{ii}}$ is a $10,240 \times 10,240$ matrix, which is impractical to invert. Hence we resort to inter-domain inducing points. However, we cannot use the usual inducing patch scheme (van der Wilk et al., 2017), as it requires access to test/train feature patches at intermediate layers, but we only have Gram matrices.

Therefore we propose a new scheme where the inducing points do not have image-like spatial structure. In particular, the Gram matrix blocks have sizes $\mathbf{G}^\ell_{\mathrm{ii}} \in \mathbb{R}^{P_{\mathrm{i}}^\ell \times P_{\mathrm{i}}^\ell}$, $\mathbf{G}^\ell_{\mathrm{it}} \in \mathbb{R}^{P_{\mathrm{i}}^\ell \times S_\ell P_{\mathrm{t}}}$ and $\mathbf{G}^\ell_{\mathrm{tt}} \in \mathbb{R}^{S_\ell P_{\mathrm{t}} \times S_\ell P_{\mathrm{t}}}$, so that the full Gram matrices are $\mathbf{G}^\ell \in \mathbb{R}^{(P_{\mathrm{i}}^\ell + S_\ell P_{\mathrm{t}}) \times (P_{\mathrm{i}}^\ell + S_\ell P_{\mathrm{t}})}$, with one row/column for each inducing point, and an additional row/column for each location in each of the train/test images.

Note that $P_i^\ell$ will be able to vary by layer in our inducing-point scheme. All kernel/Gram matrix like objects (specifically, the outputs of $\mathbf{\Gamma}(\cdot)$ and $\mathbf{\Omega}(\cdot)$) have this same structure.

We now show how we define these inducing points and compute the relevant covariances, forming the primary technical contribution of this paper. Previously, we considered only spatially structured $\mathbf{H}^{\ell-1}$ and $\mathbf{F}^\ell$. Now, we consider $\mathbf{H}^{\ell-1} \in \mathbb{R}^{(P_i^{\ell-1}+S_{\ell-1}P_t)\times N_{\ell-1}}$, formed by concatenating non-spatial $\mathbf{H}_i^{\ell-1} \in \mathbb{R}^{P_i^{\ell-1}\times N_{\ell-1}}$ (subscript "i" for inducing points) with spatially structured $\mathbf{H}_t^{\ell-1} \in \mathbb{R}^{S_{\ell-1}P_t\times N_{\ell-1}}$ (subscript "t" for test/train points). Likewise, we have $\mathbf{F}^\ell \in \mathbb{R}^{(P_i^\ell+S_\ell P_t)\times N_\ell}$ formed by combining non-spatial $\mathbf{F}_i^\ell \in \mathbb{R}^{P_i^\ell\times N_\ell}$ with spatially structured $\mathbf{F}_t^\ell \in \mathbb{R}^{S_\ell P_t\times N_\ell}$,

$$\mathbf{H}^{\ell-1} = \begin{pmatrix} \mathbf{H}_i^{\ell-1} \\ \mathbf{H}_t^{\ell-1} \end{pmatrix} \qquad\qquad \mathbf{F}^\ell = \begin{pmatrix} \mathbf{F}_i^\ell \\ \mathbf{F}_t^\ell \end{pmatrix}. \tag{14}$$

As in Eq. 10, we define $\mathbf{\Omega}^{\ell-1}$ as the normalised product of the activations with themselves. Breaking this expression into blocks (with the bottom right block $\mathbf{\Omega}_{tt}^{\ell-1}$ matching Eq. (10)), this is

$$\begin{pmatrix} \mathbf{\Omega}_{ii}^{\ell-1} & \mathbf{\Omega}_{it}^{\ell-1} \\ \mathbf{\Omega}_{ti}^{\ell-1} & \mathbf{\Omega}_{tt}^{\ell-1} \end{pmatrix} = \tfrac{1}{N_{\ell-1}} \begin{pmatrix} \mathbf{H}_i^{\ell-1}\left(\mathbf{H}_i^{\ell-1}\right)^T & \mathbf{H}_i^{\ell-1}\left(\mathbf{H}_t^{\ell-1}\right)^T \\ \mathbf{H}_t^{\ell-1}\left(\mathbf{H}_i^{\ell-1}\right)^T & \mathbf{H}_t^{\ell-1}\left(\mathbf{H}_t^{\ell-1}\right)^T \end{pmatrix}. \tag{15}$$

Note that while the features in Eq. 14 can only be realised for finite $N_\ell$, Eq. 15 is valid as $N_\ell \to \infty$. As in Eq. 13, the kernel matrix $\mathbf{K}_{\text{Conv}}(\mathbf{G}^{\ell-1})$ is formed as the covariance of features, with $\mathbf{f}_\lambda^\ell \in \mathbb{R}^{(P_i^\ell+S_\ell P_t)}$ a single feature vector / column of $\mathbf{F}^\ell$. Broken into the inducing and test/train blocks,

$$\begin{pmatrix} \mathbf{K}_{ii}^{\text{Conv}} & \mathbf{K}_{it}^{\text{Conv}} \\ \mathbf{K}_{ti}^{\text{Conv}} & \mathbf{K}_{tt}^{\text{Conv}} \end{pmatrix} = \begin{pmatrix} \mathbf{\Gamma}_{ii}(\mathbf{\Omega}_{ii}^{\ell-1}) & \mathbf{\Gamma}_{it}(\mathbf{\Omega}_{it}^{\ell-1}) \\ \mathbf{\Gamma}_{ti}(\mathbf{\Omega}_{ti}^{\ell-1}) & \mathbf{\Gamma}_{tt}(\mathbf{\Omega}_{tt}^{\ell-1}) \end{pmatrix} = \mathbb{E}\left[ \begin{pmatrix} \mathbf{f}_\lambda^{i;\ell}\left(\mathbf{f}_\lambda^{i;\ell}\right)^T & \mathbf{f}_\lambda^{i;\ell}\left(\mathbf{f}_\lambda^\ell\right)^T \\ \mathbf{f}_\lambda^\ell\left(\mathbf{f}_\lambda^{i;\ell}\right)^T & \mathbf{f}_\lambda^\ell\left(\mathbf{f}_\lambda^\ell\right)^T \end{pmatrix} \middle| \mathbf{H}^{\ell-1} \right]. \tag{16}$$

At a high-level, this approach is a natural extension of the sparse DKM procedure (Yang et al., 2023). In particular, the test/train block $\mathbf{\Gamma}_{tt}(\mathbf{\Omega}_{tt}^{\ell-1})$ remains the same as Eq. (13). However, to get concrete forms for $\mathbf{\Gamma}_{it}$ and $\mathbf{\Gamma}_{ii}$ we need to choose $\mathbf{f}_\lambda^{i;\ell}$. For this $\mathbf{f}_\lambda^{i;\ell}$ to work well, it needs to have useful, informative correlations with $\mathbf{f}_\lambda^{t;\ell}$. One way to achieve this is to define $\mathbf{F}_i^\ell$ in terms of the same weights $\mathbf{W}^\ell$ as $\mathbf{F}_t^\ell$ (from Eq. 8), thereby making them correlated. However, it is not obvious how to do this, as $\mathbf{W}^\ell$ in Eq. (8) is a convolutional filter, with spatially-structured inputs and outputs, while $\mathbf{F}_i^\ell \in \mathbb{R}^{P_i^\ell\times N}$ has no spatial structure. Our solution is to create artificial patches from linear combinations of the inducing points using a new learned parameter, $\mathbf{C}^\ell \in \mathbb{R}^{DP_i^\ell\times P_i^{\ell-1}}$,

$$F_{i,\lambda}^{i;\ell} = \sum_{d\in\mathcal{D}}\sum_{\mu=1}^N W_{d\mu,\lambda}^\ell \sum_{i'} C_{di,i'}^\ell H_{i',\mu}^{i;\ell-1}. \tag{17}$$

Intuitively, $\sum_{i'} C_{di,i'}^\ell H_{i',\mu}^{i;\ell-1}$ takes non-spatial inducing inputs, $H_{i',\mu}^{i;\ell-1}$, and converts them into a tensor with shape $DP_i^\ell \times N_{\ell-1}$, which can be understood as patches. Those patches can then be convolved with $\mathbf{W}^\ell$ to get a single output for each patch, $F_{i,\lambda}^{i;\ell}$. As a neat side effect, note that changing $P_i^\ell$ in $\mathbf{C}^\ell \in \mathbb{R}^{DP_i^\ell\times P_i^{\ell-1}}$ lets us vary the number of inducing points at each layer. The parameters $\mathbf{C}^\ell$ are learned by optimising Eq. (7). Using $\mathbf{F}_i^\ell$, we can then compute the covariances (Appendix B) to obtain $\mathbf{\Gamma}_{it}(\mathbf{\Omega}_{it}^{\ell-1})$ and $\mathbf{\Gamma}_{ii}(\mathbf{\Omega}_{ii}^{\ell-1})$ (we again emphasise that all features are IID, so the covariance is defined for an arbitrary feature $\lambda$):

$$\Gamma_{i,j}^{ii}(\mathbf{\Omega}_{ii}^{\ell-1}) = \mathbb{E}\left[F_{i,\lambda}^{i;\ell}\left(F_{j,\lambda}^{i;\ell}\right)\middle|\mathbf{H}^{\ell-1}\right] = \tfrac{1}{D}\sum_{d\in\mathcal{D}}\sum_{i'=1}^{P_i^{\ell-1}}\sum_{j'=1}^{P_i^{\ell-1}} C_{di,i'}^\ell C_{dj,j'}^\ell \Omega_{i'j'}^{ii;\ell-1} \tag{18}$$

$$\Gamma_{i,js}^{it}(\mathbf{\Omega}_{it}^{\ell-1}) = \mathbb{E}\left[F_{i,\lambda}^{i;\ell}\left(F_{js,\lambda}^{t;\ell}\right)\middle|\mathbf{H}^{\ell-1}\right] = \tfrac{1}{D}\sum_{d\in\mathcal{D}}\sum_{i'=1}^{P_i^{\ell-1}} C_{di,i'}^\ell \Omega_{i',j(s+d)}^{it;\ell-1}. \tag{19}$$

This fully defines our inducing point scheme. At the final layer, we collapse the spatial dimension (either using a linear layer or global average pooling, see Appendix E) and then perform classification using the final Gram matrix $\mathbf{G}^L$ as the covariance of a GP. For efficiency, we only store the diagonal of the test/train "tt" block at all layers, which is sufficient for IID likelihoods (Appendix E.2).

Table 1: Test accuracies (%) for model selection experiments. Base model used $\nu_\ell = 1$, Batch / Batch normalisation, Batch / Batch rescaling, Global Average Pooling, and a categorical likelihood. Bold shows values statistically similar to the maximum in each category, according to a one-tailed Welch test.

| Category | Setting | MNIST | CIFAR-10 | CIFAR-100 |
|---|---|---|---|---|
| Regularisation Strength $\nu_\ell$ | $\infty$ (inducing NNGP) | $72.19 \pm 0.11$ | $53.74 \pm 0.38$ | $30.94 \pm 0.54$ |
| | $10^4$ | $80.79 \pm 0.49$ | $52.26 \pm 0.27$ | $28.54 \pm 0.11$ |
| | $10^2$ | $96.96 \pm 0.10$ | $71.06 \pm 0.11$ | $48.23 \pm 0.11$ |
| | $10^0$ | $\mathbf{99.20 \pm 0.02}$ | $87.86 \pm 0.09$ | $64.08 \pm 0.24$ |
| | $10^{-2}$ | $\mathbf{99.16 \pm 0.05}$ | $\mathbf{89.48 \pm 0.15}$ | $\mathbf{65.64 \pm 0.20}$ |
| | $10^{-4}$ | $\mathbf{99.14 \pm 0.05}$ | $\mathbf{89.68 \pm 0.03}$ | $\mathbf{65.91 \pm 0.20}$ |
| | $0$ | $\mathbf{99.16 \pm 0.05}$ | $\mathbf{89.75 \pm 0.09}$ | $\mathbf{65.93 \pm 0.19}$ |
| Normalisation (Inducing / Train-Test) | Batch / Batch | $\mathbf{99.20 \pm 0.02}$ | $\mathbf{87.86 \pm 0.09}$ | $\mathbf{64.08 \pm 0.24}$ |
| | Batch / Location | $99.05 \pm 0.03$ | $\mathbf{87.91 \pm 0.17}$ | $\mathbf{64.05 \pm 0.27}$ |
| | Local / Image | $\mathbf{99.14 \pm 0.01}$ | $\mathbf{87.73 \pm 0.12}$ | $\mathbf{64.37 \pm 0.28}$ |
| | Local / Local | $\mathbf{99.13 \pm 0.05}$ | $\mathbf{87.54 \pm 0.09}$ | $\mathbf{63.75 \pm 0.11}$ |
| | None / None | $\mathbf{99.16 \pm 0.04}$ | $\mathbf{87.50 \pm 0.19}$ | $\mathbf{64.23 \pm 0.08}$ |
| Rescaling (Inducing / Train-Test) | Batch / Batch | $\mathbf{99.20 \pm 0.02}$ | $\mathbf{87.86 \pm 0.09}$ | $\mathbf{64.08 \pm 0.24}$ |
| | Batch / Location | $99.07 \pm 0.02$ | $\mathbf{87.98 \pm 0.16}$ | $62.72 \pm 0.15$ |
| | Local / Batch | $\mathbf{99.19 \pm 0.02}$ | $87.56 \pm 0.05$ | $\mathbf{63.92 \pm 0.13}$ |
| | Local / Location | $99.03 \pm 0.03$ | $\mathbf{87.80 \pm 0.04}$ | $\mathbf{63.30 \pm 0.43}$ |
| | Local / None | $\mathbf{99.13 \pm 0.04}$ | $\mathbf{87.75 \pm 0.04}$ | $\mathbf{63.88 \pm 0.20}$ |
| | None / None | $\mathbf{99.13 \pm 0.04}$ | $\mathbf{87.77 \pm 0.11}$ | $\mathbf{64.05 \pm 0.14}$ |
| Top Layer | Global Average Pooling | $\mathbf{99.20 \pm 0.02}$ | $\mathbf{87.86 \pm 0.09}$ | $\mathbf{64.08 \pm 0.24}$ |
| | Linear | $\mathbf{99.13 \pm 0.03}$ | $\mathbf{87.89 \pm 0.15}$ | $\mathbf{64.06 \pm 0.19}$ |
| | GAP + Mixup | $99.15 \pm 0.01$ | $\mathbf{87.69 \pm 0.10}$ | $\mathbf{64.42 \pm 0.34}$ |
| | Linear + Mixup | $99.12 \pm 0.02$ | $\mathbf{87.66 \pm 0.05}$ | $63.01 \pm 0.08$ |
| Likelihood | Gaussian | $99.05 \pm 0.01$ | $85.96 \pm 0.29$ | $26.93 \pm 0.69$ |
| | Categorical | $\mathbf{99.20 \pm 0.02}$ | $\mathbf{87.86 \pm 0.09}$ | $\mathbf{64.08 \pm 0.24}$ |

Table 2: Test accuracies (%) using different numbers of inducing points in the ResNet blocks, selecting the best hyperparameters from Table 1. Bold shows values statistically similar to the maximum in each category, according to a one-tailed Welch test. Also included is walltime per epoch.

| Ind. Points | MNIST | CIFAR-10 | CIFAR-100 | Time per epoch (s) |
|---|---|---|---|---|
| 16 / 32 / 64 | $98.99 \pm 0.06$ | $83.15 \pm 0.26$ | $53.89 \pm 0.25$ | 45 |
| 32 / 64 / 128 | $99.09 \pm 0.05$ | $86.75 \pm 0.14$ | $61.10 \pm 0.22$ | 54 |
| 64 / 128 / 256 | $99.12 \pm 0.01$ | $88.99 \pm 0.05$ | $64.31 \pm 0.19$ | 82 |
| 128 / 256 / 512 | $99.10 \pm 0.04$ | $91.00 \pm 0.16$ | $67.14 \pm 0.16$ | 155 |
| 256 / 512 / 1024 | $99.16 \pm 0.03$ | $92.17 \pm 0.10$ | $70.33 \pm 0.23$ | 420 |
| 512 / 1024 / 2048 | $\mathbf{99.26 \pm 0.02}$ | $\mathbf{92.69 \pm 0.06}$ | $\mathbf{72.05 \pm 0.23}$ | 1380 |

## 5 EXPERIMENTS

**Model variants** To fully evaluate our algorithm, it was necessary to introduce and test a variety of model variants and new techniques. In particular, we consider 5 hyperparameters/architectural choices: the regularisation strength constant $\nu$ in the objective function (Eq. 7), the normalisation and rescaling schemes (Appendix C), the top layer, i.e. linear (Appendix E.1) vs. global average pooling (Appendix E.3), and the likelihood (Appendix D.2). We vary each hyperparameter in turn, starting from a reasonable "default" model, which uses $\nu = 1$ (we use the same $\nu_\ell$ for all layers), "Batch / Batch" normalisation and "Batch / Batch" scaling (we can apply different variants to the inducing and

Table 3: Comparison of test accuracy against other kernel methods with and without learned parameters.

|  | Paper | Method | CIFAR-10 |
|---|---|---|---|
|  | This paper | DKM-DA-GAP | **92.69%** |
| Kernel methods without parameters | Novak et al. (2018) | NNGP-GAP | 77.43% |
|  | Arora et al. (2019) | NNGP-GAP | 83.75% |
|  | Lee et al. (2020) | NNGP-GAP-DA | 84.8% |
|  | Li et al. (2019) | NNGP-LAP-flip | 88.92% |
|  | Shankar et al. (2020) | Myrtle10 | 89.80% |
|  | Adlam et al. (2023) | Tuned Myrtle10 DA CG | 91.2% |
| Kernel methods with parameters | (Shi et al., 2020) | SOVI Conv GP | 68.19% |
|  | (Blomqvist et al., 2018) | Conv DGP | 75.89% |
|  | (Dutordoir et al., 2020) | Conv DGP | 76.17% |
|  | (Mairal et al., 2014) | CKN | 82.18% |
|  | (Mairal, 2016) | Sup CKN | 89.8% |

test-train blocks of the Gram matrices, see Appendix C), global average pooling, and a categorical likelihood.

We used a model architecture that mirrors the ResNet20 architecture (He et al., 2016), meaning we use the the same number of convolution layers with the same strides, we use normalisation in the same places, and we use skip connections, which compute a convex combination of the Gram matrix outputted by a block with the inputted Gram matrix, thereby allowing information to "skip" over blocks. To mirror the ReLU, we used a first-order arccos kernel (Cho & Saul, 2009) for $\Omega(\cdot)$ everywhere. Where the original ResNet20 architecture uses $\{16,32,64\}$ channels in its blocks, we initially used $\{128,256,512\}$ inducing points (i.e. we set $P_i^\ell = 128$ for layers $\ell$ contained in the first block, $P_i^\ell = 256$ in the second block, and $P_i^\ell = 512$ in the third block). This is only possible since our inducing point scheme allows $P_i^\ell$ to vary across layers, which is not true of the original sparse DKM. $\{128,256,512\}$ was chosen to balance model capacity and computational resources for the model selection experiments. The model was implemented[2] in PyTorch (Paszke et al., 2019), using double precision floating points. We optimised all parameters using Adam, with $\beta_1 = 0.8$, $\beta_2 = 0.9$, training for 100 epochs and dividing the learning rate by 10 at epochs 40 and 80, with an initial learning rate of $0.01$[3]. More careful optimisation (e.g. more epochs and less aggressive scheduling) could potentially give better results, but in the interest of total experiment time we chose this heuristic. We used data augmentation (random cropping and horizontal flips), and a batch size of 256. We also used ZCA (zero-phase component analysis) whitening, which is commonly used in kernel methods (e.g. Shankar et al., 2020; Lee et al., 2020) (for the ZCA regularisation parameter, $\epsilon$, we used 0.1).

We initialise the inducing inputs by randomly selecting patches from the training set, and then initialise the inducing Gram blocks $\mathbf{G}_{ii}^\ell$ at the NNGP case by propagating the inducing inputs through the model and setting $\mathbf{G}_{ii}^\ell$ at each layer equal to the incoming kernel matrix block $\mathbf{K}_{ii}^\ell$. The inducing output parameters $\boldsymbol{\mu}$ and $\mathbf{A}$ are initialised randomly from a Gaussian distribution (for $\mathbf{A}$ we initialise $\mathbf{L}$ as Gaussian and then compute $\mathbf{A} = \mathbf{L}\mathbf{L}^T$ to ensure $\mathbf{A}$ is positive-definite). We initialise the mixup parameters $\mathbf{C}^\ell$ randomly.

We tested on the MNIST[4] (LeCun et al., 1998), CIFAR-10, and CIFAR-100[5] (Krizhevsky & Hinton, 2009) image classification datasets. We report mean test accuracy with standard errors (over 4 random seeds) in Table 1; see Appendix F for other metrics. For each hyperparameter and dataset, we bold all settings whose performance was statistically similar to the best, using a one-tailed Welch's t-test at a 5% significance level. For the normalisation and rescaling schemes, we test certain combinations for the inducing and test-train blocks that we feel are natural; see Appendix F.2 for more details.

---

[2]Code: https://github.com/edwardmilsom/convdkmpaper
[3]The $\nu_\ell = \infty$ runs for MNIST used an initial learning rate of 0.001 due to numerical issues caused by 0.01.
[4]https://yann.lecun.com/exdb/mnist/ Licence: CC BY-SA 3.0
[5]https://cs.toronto.edu/~kriz/cifar.html Licence: Unknown

Setting $\nu = \infty$ causes the KL term in the objective to dominate, and forces $\mathbf{G}^\ell = \mathbf{K}(\mathbf{G}^{\ell-1})$, as in a standard infinite-width CNN (Novak et al., 2018; Garriga-Alonso et al., 2018). As $\nu$ controls the strength of the regularisation, performance decreases for larger values, as this causes the model to underfit. On the other hand, we expect the model to overfit if we set $\nu$ too small, and we indeed observe a degradation in test LL (Table 4) though not accuracy when we set $\nu = 0$. We find $\nu = 10^{-2}$ is the strongest regularisation statistically indistinguishable from optimal accuracy. We did not observe meaningful differences in test accuracy between any of the normalisation or rescaling schemes, though not using any normalisation or rescaling can negatively impact train performance (see Table 5). Hence we selected the next simplest scheme, "Batch / Batch", for the benchmark experiments later on. This scheme divides the Gram matrix by the mean of its diagonal elements for the normalisation, and multiplies each block by a learned scalar for the rescaling. We saw little difference between the top layers, so we settled on global average pooling for later experiments, reflecting common practice in deep learning and NNGPs (e.g. Novak et al., 2018). We also tried adding inducing point "mixups" similar to the mechanism in our convolutional layers, but this seemed to have no benefit. Using a categorical likelihood provides moderate advantages for CIFAR-10 and MNIST, which have 10 classes, and dramatic improvements for CIFAR-100, which has 100 classes.

**Benchmark performance.** Using what we learned from the model selection experiments, we ran a set of experiments with $\nu_\ell = 10^{-2}$, "Batch / Batch" normalisation and rescaling, global average pooling, and a categorical likelihood. In these experiments, we varied the number of inducing points in each block from $\{16, 32, 64\}$ (in the first, middle, and last blocks) up to $\{512, 1024, 2048\}$. We trained for 200 epochs, reducing the learning rate by a factor of 10 at epochs 80 and 160, starting from an initial learning rate of 0.01. Mean test accuracies are given in Table 2 with other metrics in Appendix F. As expected, increasing the number of inducing points increases observed test accuracy, with diminishing returns for MNIST, moderate improvements for CIFAR-10, and considerable improvements for CIFAR-100. Our largest model, with 2048 inducing points in the last block, achieved 99.3% test accuracy on MNIST, 92.7% on CIFAR-10, and 72% on CIFAR-100. Training this model on CIFAR-10/100 took around 77 hours on a single NVIDIA A100, but we emphasise that we used double precision floating points for simplicity of implementation, and with some care taken to preserve numerical stability, this could be dramatically sped up using single precision arithmetic.

**Comparisons against other models.** In Table 3, we compare our best model against other pure kernel methods, including those without parameters like NNGP and NTK, and those with parameters like convolutional kernel networks. Our model outperforms even the best pure kernel methods (Adlam et al., 2023; Shankar et al., 2020; Li et al., 2019; Mairal, 2016). These models are not directly comparable to ours, as we developed an efficient inducing point scheme, whereas NNGP/NTK methods usually compute the full training kernel matrix. Benchmarks from Google's Neural Tangents library[6] found that the most efficient Myrtle-5/7/10 kernels from Shankar et al. (2020) took 316/330/508 GPU hours for the full 60 000 CIFAR-10 dataset on an Nvidia V100, which is around one order of magnitude more than our training time. Overall, we have narrowed (but not entirely eliminated) the gap to NNs by improving the performance of kernel methods.

## 6 CONCLUSION AND LIMITATIONS

We have developed convolutional DKMs, and an efficient inter-domain inducing point approximation scheme. We performed a comprehensive set of experiments to investigate the effect of different architectural choices, before selecting the final architecture to compare against existing methods. Our best models obtained 99% test accuracy on MNIST, 92.7% on CIFAR-10 and 72% on CIFAR-100. This is state-of-the-art for kernel-methods / infinite NNs, bringing theoretical approaches closer in performance to neural networks, and there is considerable room for development to close the gap even further. One limitation of our work is that we did not use high resolution image datasets like ImageNet (Deng et al., 2009) as we have not yet developed the efficient tricks (such as lower/mixed precision floating point arithmetic, more efficient memory utilization, and numerical methods for e.g. inverting matrices) to enable these larger datasets. It may be possible to adapt existing ideas from the kernel literature (Adlam et al. (2023) and Maddox et al. (2022), for example) to address some of these issues, in addition to leveraging multi-GPU setups, but we leave this to future work.

---

[6]`github.com/google/neural-tangents`

## 7 ACKNOWLEDGEMENTS

Edward Milsom and Ben Anson are funded by the Engineering and Physical Sciences Research Council via the COMPASS Centre for Doctoral Training at the University of Bristol. This work was carried out using the computational facilities of the Advanced Computing Research Centre, University of Bristol - `http://www.bris.ac.uk/acrc/`. We would like to thank Dr Stewart for funding compute resources used in this project.

## 8 REPRODUCIBILITY STATEMENT

The experiments in this paper may be reproduced using the code and instructions given in the following publicly available GitHub repo: `https://github.com/edwardmilsom/convdkmpaper`.

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

## A    DEEP KERNEL MACHINES

In this appendix we provide a more detailed introduction to deep kernel machines. Deep kernel machines (DKMs) are a deep generalisation of kernel methods that arise when considering a particular infinite limit of deep Gaussian processes (DGPs), called the Bayesian representation learning limit Yang et al. (2023). The original motivation for this limit was to study representation learning in deep models e.g. neural networks, but it turned out that when applied to DGPs, the resulting posteriors are exactly multivariate Gaussians, and hence we obtain a practical algorithm that can be used for supervised learning. It is a deep generalisation of kernel methods in the sense that the model has Gram matrices rather than features at the intermediate layers. These Gram matrices are optimised by maximising the DKM objective; the derivation of the objective and reasoning behind needing to optimise Gram matrices will be elucidated in the rest of this appendix.

The simplest way to explain DKMs is to walk through the steps Yang et al. (2023) took that led to the Bayesian representation learning limit, and more specifically how it applies to deep Gaussian processes.

### A.1    DEEP GAUSSIAN PROCESSES

We start by defining a deep Gaussian process (DGP). A DGP maps inputs $\mathbf{X} \in \mathbb{R}^{P \times N_0}$ to outputs $\mathbf{Y} \in \mathbb{R}^{P \times N_{L+1}}$ using $L$ intermediate layers of features $\mathbf{F}^\ell \in \mathbb{R}^{P \times N_\ell}$. Here $P$ is the number of input points, and $N_\ell$ is the number of features at layer $\ell$, so that $N_0$ is the number of input features, and $N_{L+1}$ is the number of output features. Writing $\mathbf{F}^\ell$ and $\mathbf{Y}$ as stacks of vectors

$$\mathbf{F}^\ell = (\mathbf{f}_1^\ell \quad \mathbf{f}_2^\ell \quad \cdots \quad \mathbf{f}_{N_\ell}^\ell) \tag{20a}$$

$$\mathbf{Y} = (\mathbf{y}_1 \quad \mathbf{y}_2 \quad \cdots \quad \mathbf{y}_{\nu_{L+1}}), \tag{20b}$$

where $\mathbf{f}_\lambda^\ell \in \mathbb{R}^P$ gives the values of a particular feature for all $P$ input points and $\mathbf{y}_\lambda \in \mathbb{R}^P$ gives the value of a particular output for all $P$ input points. The features $\mathbf{F}_1, \ldots, \mathbf{F}_L$ are sampled from a Gaussian process whose covariance depends on the previous layer's features,

$$\mathrm{P}\left(\mathbf{F}^\ell | \mathbf{F}^{\ell-1}\right) = \prod_{\lambda=1}^{\nu_\ell} \mathcal{N}\left(\mathbf{f}_\lambda^\ell; \mathbf{0}, \mathbf{K}(\mathbf{G}(\mathbf{F}^{\ell-1}))\right). \tag{21}$$

Here, we compute the kernel by first computing the Gram matrix,

$$\mathbf{G}^\ell = \mathbf{G}(\mathbf{F}^\ell) = \frac{1}{N_\ell} \sum_{\lambda=1}^{N_\ell} \mathbf{f}_\lambda^\ell (\mathbf{f}_\lambda^\ell)^T = \frac{1}{N_\ell} \mathbf{F}^\ell (\mathbf{F}^\ell)^T. \tag{22}$$

then use a kernel that can be computed directly from the Gram matrix (Eq. (3); also see Aitchison et al. 2021). To see why this works for isotropic kernels that depend only on distance, see Eq. (2). For regression the outputs are also sampled from a Gaussian process,

$$\mathrm{P}\left(\mathbf{Y} | \mathbf{F}^L\right) = \prod_{\lambda=1}^{\nu_{L+1}} \mathcal{N}\left(\mathbf{y}_\lambda; \mathbf{0}, \mathbf{K}(\mathbf{G}(\mathbf{F}^L)) + \sigma^2 \mathbf{I}\right). \tag{23}$$

Other likelihoods could be used. For example, in this paper we do classification using a categorical likelihood with softmax probabilities.

## A.2 DGPs written in terms of Gram matrices

We wish to consider the DGP analogue of the standard neural network Gaussian process (NNGP) limit, which takes the intermediate layers $\ell \in \{1, \ldots, L\}$ to have infinitely-many features (number of features here is an analogue of layer width in a neural network). Of course, we cannot work with matrices $\mathbf{F}^\ell \in \mathbb{R}^{P \times N_\ell}$ with $N_\ell \to \infty$, so we will instead work with Gram matrices that represent the inner product between features (Eq. 22). We can then rewrite the DGP as

$$\mathrm{P}\left(\mathbf{F}^\ell | \mathbf{G}^{\ell-1}\right) = \prod_{\lambda=1}^{N_\ell} \mathcal{N}\left(\mathbf{f}_\lambda^\ell; \mathbf{0}, \mathbf{K}(\mathbf{G}^{\ell-1})\right) \tag{24a}$$

$$\mathrm{P}\left(\mathbf{G}^\ell | \mathbf{F}^\ell\right) = \delta\left(\mathbf{G}^\ell - \mathbf{G}(\mathbf{F}^\ell)\right) \tag{24b}$$

$$\mathrm{P}\left(\mathbf{Y} | \mathbf{G}^L\right) = \prod_{\lambda=1}^{\nu_{L+1}} \mathcal{N}\left(\mathbf{y}_\lambda; \mathbf{0}, \mathbf{K}(\mathbf{G}^L) + \sigma^2 \mathbf{I}\right). \tag{24c}$$

We consider a regression likelihood for concreteness, but, again, other likelihoods can be used. Here, we have written $\mathrm{P}\left(\mathbf{G}^\ell | \mathbf{F}^\ell\right)$ using the Dirac-delta, as $\mathbf{G}^\ell$ is a deterministic function of $\mathbf{F}^\ell$ (Eq. 22).

All that remains to prepare the model for taking the infinite limit is to integrate out $\mathbf{F}^\ell$ from the intermediate layers (note that the likelihood already depends on $\mathbf{G}^L$ alone):

$$\mathrm{P}\left(\mathbf{G}^\ell | \mathbf{G}^{\ell-1}\right) = \int \mathrm{P}\left(\mathbf{G}^\ell | \mathbf{F}^\ell\right) \mathrm{P}\left(\mathbf{F}^\ell | \mathbf{G}^{\ell-1}\right) d\mathbf{F}^\ell. \tag{25}$$

It turns out that in the case of the DGP model, Eq. 25 is actually a Wishart distribution (one can see this by noting that 22 is the outer product of IID multivariate Gaussian vectors, which matches the definition of the Wishart),

$$\mathrm{P}\left(\mathbf{G}^\ell | \mathbf{G}^{\ell-1}\right) = \mathrm{Wishart}\left(\mathbf{G}^\ell; \frac{1}{N_\ell} \mathbf{K}(\mathbf{G}^{\ell-1}), N_\ell\right) \tag{26a}$$

$$\log \mathrm{P}\left(\mathbf{G}^\ell | \mathbf{G}^{\ell-1}\right) = \frac{N_\ell - P - 1}{2} \log \left|\mathbf{G}^\ell\right| - \frac{N_\ell}{2} \log \left|\mathbf{K}(\mathbf{G}^{\ell-1})\right| - \frac{N_\ell}{2} \mathrm{Tr}\left(\mathbf{K}^{-1}(\mathbf{G}^{\ell-1})\mathbf{G}^\ell\right) + \alpha_\ell \tag{26b}$$

where,

$$\alpha_\ell = -\frac{N_\ell P}{2} \log 2 + \frac{N_\ell P}{2} \log N_\ell - \log \Gamma_P\left(\frac{N_l}{2}\right), \tag{27}$$

is constant w.r.t. all $\mathbf{G}^\ell$ and $\Gamma_P$ is the multivariate Gamma function. Note that the distribution (Eq. 26a) is valid for any $N_\ell \geq 1$, but the log density in Eq. (26b) is valid only in the full-rank case where $N_\ell \geq P$ (though this restriction does not cause problems because we will take $N_\ell \to \infty$).

## A.3 Standard infinite-width limit

We now consider the DGP analogue of the standard NNGP infinite limit, that is we will take the intermediate layers to have infinite width / infinitely-many features:

$$N_\ell = N\nu_\ell \quad \text{for} \quad \ell \in \{1, \ldots, L\} \quad \text{with} \quad N \to \infty. \tag{28}$$

The log-posterior for the DGP written in terms of Gram matrices is

$$\log P\left(\mathbf{G}^1, \ldots, \mathbf{G}^L | \mathbf{X}, \mathbf{Y}\right) = \log P\left(\mathbf{Y}|\mathbf{G}^L\right) + \sum_{\ell=1}^{L} \log P\left(\mathbf{G}^\ell | \mathbf{G}^{\ell-1}\right) + \text{const,} \qquad (29)$$

where we emphasise that $\mathbf{G}^1, \ldots, \mathbf{G}^L$ are $P \times P$ matrix arguments chosen by us to specify where we wish to evaluate the log-posterior, and so are not affected by any infinite-width limits we take. That said, the actual value of $\log P\left(\mathbf{G}^\ell | \mathbf{G}^{\ell-1}\right)$ is affected by the limit. In particular, in the infinite limit specified in Eq. (28), the log-prior term $\log P\left(\mathbf{G}^\ell | \mathbf{G}^{\ell-1}\right)$ (Eq. 26) scales with $N_\ell$ and therefore $N$, so we must divide by $N$ to obtain a finite limit:

$$\lim_{N \to \infty} \tfrac{1}{N} \log P\left(\mathbf{G}^\ell | \mathbf{G}^{\ell-1}\right) = \tfrac{\nu_\ell}{2} \left( \log \left| \mathbf{K}^{-1}(\mathbf{G}^{\ell-1})\mathbf{G}^\ell \right| - \text{Tr}\left(\mathbf{K}^{-1}(\mathbf{G}^{\ell-1})\mathbf{G}^\ell\right) \right) + \lim_{N \to \infty} \tfrac{\alpha_\ell}{N}$$

$$= -\nu_\ell \, \text{D}_{\text{KL}}\left( \mathcal{N}\left(\mathbf{0}, \mathbf{G}^\ell\right) \big\| \mathcal{N}\left(\mathbf{0}, \mathbf{K}(\mathbf{G}^{\ell-1})\right) \right) + \text{const.} \qquad (30)$$

where $\lim_{N \to \infty} \alpha_\ell / N$ is given in Yang et al. (2023). Note that this limit has been written as a KL-divergence between two Gaussians. By contrast, the log-likelihood term $\log P\left(\mathbf{Y}|\mathbf{G}^L\right)$ (Eq. 24c) is constant w.r.t. $N$, so will vanish if we divide by $N$ in the limit. Putting this all together, the limiting log-posterior (divided by $N$ to remain finite) is

$$\lim_{N \to \infty} \tfrac{1}{N} \log P\left(\mathbf{G}^1, \ldots, \mathbf{G}^L | \mathbf{X}, \mathbf{Y}\right) = -\sum_{\ell=1}^{L} \nu_\ell \, \text{D}_{\text{KL}}\left( \mathcal{N}\left(\mathbf{0}, \mathbf{G}^\ell\right) \big\| \mathcal{N}\left(\mathbf{0}, \mathbf{K}(\mathbf{G}^{\ell-1})\right) \right) + \text{const.}$$
$$(31)$$

Now, as this log-posterior scales with $N$, the posterior itself converges to a point distribution at its global maximum $(\mathbf{G}^1)^*, \ldots, (\mathbf{G}^L)^*$ (see Yang et al. (2023) for a more rigorous discussion of weak convergence)

$$\lim_{N \to \infty} P\left(\mathbf{G}^1, \ldots, \mathbf{G}^L | \mathbf{X}, \mathbf{Y}\right) = \prod_{\ell=1}^{L} \delta\left(\mathbf{G}^\ell - (\mathbf{G}^\ell)^*\right). \qquad (32)$$

From Eq. (31), it is clear that the unique global maximum is obtained when $(\mathbf{G}^\ell)^* = \mathbf{K}((\mathbf{G}^{\ell-1})^*)$, since this minimises the KL divergence. In other words, we can compute the maximum recursively by applying the kernel repeatedly. Importantly, this means that the limiting posterior has no dependence on $\mathbf{Y}$, our training targets, and so it is not possible for representation learning to occur. This is a big problem, since the success of modern deep learning hinges on its ability to flexibly learn good representations from data.

## A.4 The Bayesian representation learning limit and DKMs

Representation learning was lost because the likelihood term in the normalised posterior vanished in the infinite limit, whilst the prior term did not. The Bayesian representation learning limit (Yang et al., 2023) retains representation learning by sending the number of output features to infinity, in addition to the intermediate features:

$$N_\ell = N \nu_\ell \quad \text{for} \quad \ell \in \{1, \ldots, L+1\} \quad \text{with} \quad N \to \infty. \qquad (33)$$

Note that the only difference between this limit and Eq. 33 is that we include layer $L+1$ (the output layer) in our limit. This defines a valid probabilistic model with a well-defined posterior using the same prior as before (Eq. 25), and a likelihood assuming each output channel is IID:

$$P\left(\tilde{\mathbf{Y}}|\mathbf{G}^L\right) = \prod_{\lambda=1}^{N_{L+1}} \mathcal{N}\left(\tilde{\mathbf{y}}_\lambda; \mathbf{0}, \mathbf{K}(\mathbf{G}^L) + \sigma^2 \mathbf{I}\right). \qquad (34)$$

where, to distinguish from the usual likelihood (Eq. 24c) $\tilde{\mathbf{Y}} \in \mathbb{R}^{P \times N_{L+1}}$ is infinite width (Eq. 33) whereas the usual DGP data, $\mathbf{Y} \in \mathbb{R}^{P \times \nu_{L+1}}$, is finite width.

In practice, infinite-width data does not usually exist. To apply the DKM to finite-width data $\mathbf{Y} \in \mathbb{R}^{P \times \nu_{L+1}}$, we let $\tilde{\mathbf{Y}}$ be formed from $N$ copies of $\mathbf{Y}$ concatenated together, i.e.

$$\tilde{\mathbf{Y}} = \begin{pmatrix} \mathbf{Y} & \cdots & \mathbf{Y} \end{pmatrix}. \qquad (35)$$

Since we assumed the output channels to be IID (Eq. 34), the log-likelihood now scales with $N$:

$$\log P\left(\tilde{\mathbf{Y}}|\mathbf{G}^L\right) = N \log P\left(\mathbf{Y}|\mathbf{G}^L\right). \qquad (36)$$

Computing the log-posterior as before, we therefore find that the likelihood no longer vanishes in the infinite limit, thus allowing representation learning:

$$
\mathcal{L}(\mathbf{G}^1, \ldots, \mathbf{G}^L) = \lim_{N \to \infty} \frac{1}{N} \log \mathrm{P}\left(\mathbf{G}^1, \ldots, \mathbf{G}^L | \mathbf{X}, \tilde{\mathbf{Y}}\right) + \text{const},
$$
$$
= \log \mathrm{P}\left(\mathbf{Y}|\mathbf{G}^L\right) - \sum_{\ell=1}^{L} \nu_\ell \, \mathrm{D}_{\mathrm{KL}}\left(\mathcal{N}\left(\mathbf{0}, \mathbf{G}^\ell\right) \middle\| \mathcal{N}\left(\mathbf{0}, \mathbf{K}(\mathbf{G}^{\ell-1})\right)\right). \tag{37}
$$

The limiting normalised log-posterior, $\mathcal{L}(\mathbf{G}^1, \ldots, \mathbf{G}^L)$, forms the "DKM objective". As before, assuming the global maximum of this expression is unique, then the posterior is a point distribution around that maximum (Eq. 32). However, unlike before, we cannot simply compute the maximum by recursively applying the kernel function to our data, since the maximum now also depends on the likelihood term. Instead, we must optimise the Gram matrices using e.g. gradient ascent, and there is no guarantee anymore that the maximum is unique.

Yang et al. (2023) use the DKM objective in practice in fully-connected settings, and call the resulting approach a "deep kernel machine". As already mentioned, the limiting log-posterior of the Gram matrices in this context is dubbed the "DKM objective" as we maximise it to fit the Gram matrices in our model. Once the Gram matrices have been optimised according to the DKM objective using the training data, one can perform prediction on new test points using a method similar to that in DGPs (see Yang et al. 2023 for details).

## B    FULL DERIVATIONS FOR THE KERNEL CONVOLUTION

Here we give the full derivations for the covariances used to define the kernel convolution in the main text. For the full-rank (no sparse inducing point scheme) case, the reader can skip straight to Section B.3. Sections B.1 and B.2 only concern the inducing point scheme.

### B.1    INDUCING BLOCK

For the inducing ii'th component,

$$
\Gamma_{ij}^{\mathrm{ii};\ell}(\mathbf{\Omega}_{\mathrm{ii}}^{\ell-1}) = \mathbb{E}\left[F_{i,\lambda}^{\mathrm{i};\ell} F_{j,\lambda}^{\mathrm{i};\ell}\right]. \tag{38}
$$

Substituting the definition of $F_{i,\lambda}^{\mathrm{i};\ell}$ (Eq. 17),

$$
= \mathbb{E}\left[\sum_{\mu d i'} W_{d\mu,\lambda}^\ell C_{di,i'}^\ell H_{i',\mu}^{\mathrm{i};\ell-1} \sum_{\nu d' j'} W_{d'\nu,\lambda}^\ell C_{d'j,j'}^\ell H_{j',\nu}^{\mathrm{i};\ell-1}\right]. \tag{39}
$$

The only random variables are the weights,

$$
= \sum_{\mu\nu, dd', i'j'} C_{di,i'}^\ell H_{i',\mu}^{\mathrm{i};\ell-1} C_{d'j,j'}^\ell H_{j',\nu}^{\mathrm{i};\ell-1} \, \mathbb{E}\left[W_{d\mu,\lambda}^\ell W_{d'\nu,\lambda}^\ell\right] \tag{40}
$$

As the weights are IID with zero mean and variance $1/DN_{\ell-1}$ (Eq. 9),

$$
= \sum_{\mu\nu, dd', i'j'} C_{di,i'}^\ell H_{i',\mu}^{\mathrm{i};\ell-1} C_{d'j,j'}^\ell H_{j',\nu}^{\mathrm{i};\ell-1} \frac{1}{DN_{\ell-1}} \delta_{\mu,\nu} \delta_{d,d'}, \tag{41}
$$

where $\delta_{\mu,\nu}$ is the Kronecker-delta which is 1 when $\mu = \nu$ and zero otherwise. The Kronecker-deltas pick out elements of the sum,

$$
= \frac{1}{DN_{\ell-1}} \sum_{\mu, d, i'j'} C_{di,i'}^\ell H_{i',\mu}^{\mathrm{i};\ell-1} C_{dj,j'}^\ell H_{j',\mu}^{\mathrm{i};\ell-1}. \tag{42}
$$

Rearrange to bring together the multiplication of previous-layer activations, $\mathbf{H}^{\ell-1}$,

$$
= \frac{1}{D} \sum_{d, i'j'} C_{di,i'}^\ell C_{dj,j'}^\ell \frac{1}{N_{\ell-1}} \sum_\mu H_{i',\mu}^{\mathrm{i};\ell-1} H_{j',\mu}^{\mathrm{i};\ell-1}. \tag{43}
$$

And remembering the definition of $\Omega_{i'j'}^{\mathrm{ii};\ell-1}$ (Eq. 10),

$$= \frac{1}{D} \sum_{d,i'j'} C_{di,i'}^{\ell} C_{dj,j'}^{\ell} \Omega_{i'j'}^{\mathrm{ii};\ell-1} \tag{44}$$

In practice, we implement this by averaging the result of $D$ batched matrix multiplications,

$$\mathbf{\Gamma}_{\mathrm{ii}}^{\ell}(\mathbf{\Omega}_{\mathrm{ii}}^{\ell-1}) = \frac{1}{D} \sum_d \mathbf{C}_d^{\ell} \mathbf{\Omega}_{\mathrm{ii}}^{\ell-1} \left(\mathbf{C}_d^{\ell}\right)^T \tag{45}$$

## B.2 INDUCING-TEST/TRAIN BLOCK

We follow the same process for the inducing-test/train it'th component,

$$\Gamma_{ij}^{\mathrm{it};\ell}(\mathbf{\Omega}_{\mathrm{it}}^{\ell-1}) = \mathbb{E}\left[F_{i,\lambda}^{\mathrm{i};\ell} F_{js,\lambda}^{\mathrm{t};\ell}\right] \tag{46}$$

$$= \mathbb{E}\left[\sum_{d\mu i'} W_{d\mu,\lambda}^{\ell} C_{di,i'}^{\ell} H_{i',\mu}^{\mathrm{i};\ell-1} \sum_{\nu d'} H_{j,(s+d')\nu}^{\ell-1} W_{d'\nu,\lambda}^{\ell}\right] \tag{47}$$

$$= \sum_{dd',\mu\nu,i'} C_{di,i'}^{\ell} H_{i',\mu}^{\mathrm{i};\ell-1} H_{j,(s+d')\nu}^{\mathrm{t};\ell-1} \; \mathbb{E}\left[W_{d\mu,\lambda}^{\ell} W_{d'\nu,\lambda}^{\ell}\right] \tag{48}$$

$$= \sum_{dd',\mu\nu,i'} C_{di,i'}^{\ell} H_{i',\mu}^{\mathrm{i};\ell-1} H_{j,(s+d')\nu}^{\mathrm{t};\ell-1} \frac{1}{DN_{\ell-1}}\delta_{\mu,\nu}\delta_{d,d'} \tag{49}$$

$$= \frac{1}{DN_{\ell-1}} \sum_{d,\mu,i'} C_{di,i'}^{\ell} H_{i',\mu}^{\mathrm{i};\ell-1} H_{j,(s+d)\mu}^{\mathrm{t};\ell-1} \tag{50}$$

$$= \frac{1}{D} \sum_{d,i'} C_{di,i'}^{\ell} \Omega_{i',j(s+d)}^{\mathrm{it};\ell-1} \tag{51}$$

In practice, this can implemented by a standard convolution operation, treating $C_{di,i'}$ as the "weights", and $\Omega_{i',j(s+d)}^{\mathrm{it};\ell-1}$ as the "inputs".

## B.3 TEST/TRAIN BLOCK

Again, we follow the same process for the test/train tt'th component. Note that $\mathbf{\Gamma}$ does not require a superscript $\ell$ here since the function is the same no matter what layer it is operating on, as there are no weights $\mathbf{C}$ when we are dealing with just test/train points.

$$\Gamma_{ir,js}^{\mathrm{tt}}(\mathbf{\Omega}_{\mathrm{tt}}^{\ell-1}) = \mathbb{E}\left[F_{i,r\lambda}^{\mathrm{t};\ell} F_{j,s\lambda}^{\mathrm{t};\ell}\right]. \tag{52}$$

$$= \mathbb{E}\left[\sum_{d\mu} H_{i,(r+d)\mu}^{\mathrm{t};\ell-1} W_{d\mu,\lambda}^{\ell} \sum_{d'\nu} H_{j,(s+d')\nu}^{\mathrm{t};\ell-1} W_{d\nu,\lambda}^{\ell}\right] \tag{53}$$

$$= \sum_{dd',\mu\nu} H_{i,r'\mu}^{\mathrm{t};\ell-1} H_{j,s'\nu}^{\mathrm{t};\ell-1} \mathbb{E}\left[W_{d\mu,\lambda}^{\ell} W_{d'\nu,\lambda}^{\ell}\right] \tag{54}$$

$$= \sum_{dd',\mu\nu} H_{i,(r+d)\mu}^{\mathrm{t};\ell-1} H_{j,(s+d')\nu}^{\mathrm{t};\ell-1} \frac{1}{DN_{\ell-1}}\delta_{\mu,\nu}\delta_{d,d'} \tag{55}$$

$$= \frac{1}{DN_{\ell-1}} \sum_{d\mu} H_{i,(r+d)\mu}^{\mathrm{t};\ell-1} H_{j,(s+d)\mu}^{\mathrm{t};\ell-1} \tag{56}$$

$$= \frac{1}{D} \sum_d \Omega_{i(r+d),j(s+d)}^{\mathrm{tt};\ell-1}. \tag{57}$$

Remember that we end up only needing the diagonal,

$$\Gamma_{ir,ir}^{\mathrm{tt}}(\mathbf{\Omega}_{\mathrm{tt}}^{\ell-1}) = \frac{1}{D} \sum_d \Omega_{i(r+d),i(r+d)}^{\mathrm{tt};\ell-1}, \tag{58}$$

and this operation can be implemented efficiently as average-pooling.

## C  Normalisation and scaling for Gram matrices

We develop various analogues to batch/layer normalisation (using the terms to loosely describe any technique for normalising activations) that can be used on Gram matrices in order to improve the performance of our model. Since we do not have explicit features, there is some flexibility in how to do this. Here we describe the variants we tested and comment on them.

We think of batchnorm as consisting of two components: normalisation and scaling. Normalisation is parameter-free, and is analogous to dividing by feature variances in standard batchnorm. Scaling has parameters, and is analogous to multiplying the features by a learned scalar in standard batchnorm. In our setting, we can apply different normalisation and scaling schemes to the inducing and test/train points: so we end up with four choices: inducing normalisation, test/train normalisation, inducing scaling and test/train scaling.

All of our batchnorm-like schemes have the following form,

$$N_{ij}^{\text{ii}}(\mathbf{G}) = G_{ij}^{\text{ii}} \frac{\psi_i^2}{a_i(\mathbf{G_{ii}}) a_j(\mathbf{G_{ii}})} \tag{59}$$

$$N_{i,js}^{\text{it}}(\mathbf{G}) = G_{i,js}^{\text{it}} \frac{\psi_i \Psi_s}{a_i(\mathbf{G_{ii}}) A_{js}(\mathbf{G_{tt}})} \tag{60}$$

$$N_{ir,ir}^{\text{tt}}(\mathbf{G}) = G_{ir,ir}^{\text{tt}} \frac{\Psi_r^2}{A_{ir}^2(\mathbf{G_{tt}})} \tag{61}$$

where $N(\mathbf{G})$ is our normalized Gram matrix. Here, $\psi_i$ is the learned parameter that represents the inducing scale and $\Psi_r$ is the learned, potentially location-dependent parameter that represents the test/train scaling. Likewise, $a_i(\mathbf{G}_{ii})$ and $A_{ir}(\mathbf{G}_{tt})$ represent the normalisation terms for inducing and test/train points respectively. In the subsequent sections, we explore potential choices for $a_i(\mathbf{G}_{ii})$ and $A_{ir}(\mathbf{G}_{tt})$ corresponding to different normalisation schemes, and choices for $\psi_i$ and $\Psi_r$ corresponding to different scaling schemes.

### C.1  Inducing normalisation

**No inducing normalisation**. We can turn off inducing normalisation by choosing,

$$a_i(\mathbf{G_{ii}}) = 1. \tag{62a}$$

**Kernel Batch inducing normalisation** is,

$$a_i(\mathbf{G_{ii}}) = a^{\text{norm}}(\mathbf{G_{ii}}) = \sqrt{\frac{1}{P_i} \sum_{j=1}^{P_i} G_{jj}^{\text{ii}}}. \tag{62b}$$

**Kernel Local inducing normalisation** is,

$$a_i(\mathbf{G_{ii}}) = a_i^{\text{norm}}(\mathbf{G_{ii}}) = \sqrt{G_{ii}^{\text{ii}}}. \tag{62c}$$

### C.2  Test/train normalisation

There are a much richer set of choices for how to normalize the test/train points, as they are spatially structured.

**No test/train normalisation.** We can turn off normalisation by choosing,

$$A_{ir}(\mathbf{G_{tt}}) = 1. \tag{63a}$$

**Kernel Batch normalisation** averages the diagonal of $\mathbf{G_{tt}}$ over all spatial locations in all images to get a single scalar normalizer,

$$A_{ir}(\mathbf{G_{tt}}) = A^{\text{batch}}(\mathbf{G_{tt}}) = \sqrt{\frac{1}{SP_t} \sum_{j=1}^{P_t} \sum_{s=1}^{S} G_{js,js}^{\text{tt}}}. \tag{63b}$$

**Kernel Image normalisation** averages the diagonal of $\mathbf{G_{tt}}$ over all spatial locations for separate images to get a different normalizer for each image,

$$A_{ir}(\mathbf{G_{tt}}) = A_i^{\text{image}}(\mathbf{G_{tt}}) = \sqrt{\frac{1}{S} \sum_{s=1}^{S} G_{is,is}^{\text{tt}}}. \tag{63c}$$

**Kernel Location normalisation**, averages the diagonal of $\mathbf{G}_{\text{tt}}$ over all images for separate spatial locations to get a different normalizer for each spatial location,

$$A_{ir}(\mathbf{G}_{\text{tt}}) = A_r^{\text{location}}(\mathbf{G}_{\text{tt}}) = \sqrt{\tfrac{1}{P_t}\sum_{j=1}^{P_t} G_{jr,jr}^{\text{tt}}}. \tag{63d}$$

**Kernel Local normalisation** does not average at all, so the normalizer is just the square root of the diagonal of $\mathbf{G}_{\text{tt}}$,

$$A_{ir}(\mathbf{G}_{\text{tt}}) = \sqrt{G_{ir,ir}^{\text{tt}}}. \tag{63e}$$

### C.3 INDUCING SCALING

Now, there are three choices for the inducing scaling.

**No inducing scaling**. We can turn off inducing scaling by choosing,

$$\psi_i = 1. \tag{64a}$$

**Kernel Batch inducing scaling**, in which there is a single scale applied to all inducing points,

$$\psi_i = \psi^{\text{const}}. \tag{64b}$$

**Kernel Local inducing scaling**, in which there is a different scale applied to each inducing point,

$$\psi_i = \psi_i^{\text{ind}} \tag{64c}$$

### C.4 TEST/TRAIN SCALING

There are three choices for the test/train scaling.

**No test/train scaling**. We can turn off test/train scaling by choosing,

$$\Psi_r = 1. \tag{65a}$$

**Kernel Batch test/train scaling** We can choose the test/train scaling to be a constant across spatial locations,

$$\Psi_r = \Psi^{\text{const}}. \tag{65b}$$

**Kernel Location scaling** We can choose the test/train scaling to be different across spatial locations, in which case we have,

$$\Psi_r = \Psi_r^{\text{location}}. \tag{65c}$$

Note that we never apply scaling image-wise, since the batch changes every iteration, so this would be nonsensical. On the other hand, spatial location carries meaning across all images, so scaling by location makes sense.

## D COMPUTING THE GP-ELBO TERM

### D.1 AMENDED DKM OBJECTIVE

In the original deep kernel machine paper (Yang et al., 2023), the authors proposed a sparse variational inducing point scheme for DKMs, roughly based on similar work in the DGP literature by Salimbeni et al. (2018). However, rather than taking the usual approach of defining an approximate posterior over the inducing outputs $\mathbf{F}^{L+1}$, they simply treat them as variational parameters. In this paper, we take the more usual approach of defining an approximate posterior over our inducing outputs, which we take to be Gaussian:

$$Q(\mathbf{F}_i^{L+1}) = \prod_{\lambda=1}^{N_{L+1}} \mathcal{N}(\mathbf{f}_\lambda^{i;L+1}; \boldsymbol{\mu}_\lambda, \mathbf{A}_\lambda). \tag{66}$$

Here, $N_{L+1}$ is the (finite) dimension of the top layer, and $\boldsymbol{\mu}_\lambda, \mathbf{A}_\lambda$ are variational parameters for each channel $\lambda$ of the output. This modifies the resulting objective function slightly, but we believe the

resulting formula is actually more natural. Following the derivation of the inducing point scheme exactly as in Appendix I of Yang et al. (2023), we find that the new sparse objective is

$$
\begin{aligned}
&\mathcal{L}_{\mathrm{ind}}(\mathbf{F}_{\mathrm{i}}^{L+1}, \mathbf{G}_{\mathrm{ii}}^{1}, \dots, \mathbf{G}_{\mathrm{ii}}^{L}) \\
&= \lim_{N \to \infty} \tfrac{1}{N} \mathrm{ELBO}(\mathbf{F}_{\mathrm{i}}^{L+1}, \mathbf{G}_{\mathrm{ii}}^{1}, \dots, \mathbf{G}_{\mathrm{ii}}^{L}) \\
&= \underbrace{\mathbb{E}_{\mathrm{Q}}\left[\log \mathrm{P}\left(\mathbf{Y}_{\mathrm{t}}|\mathbf{F}_{\mathrm{t}}^{L+1}\right)\right] - \sum_{\lambda=1}^{N_{L+1}} \mathrm{D}_{\mathrm{KL}}\left(\mathcal{N}\left(\boldsymbol{\mu}_{\lambda}, \mathbf{A}_{\lambda}\right)\|\mathcal{N}\left(\mathbf{0}, \mathbf{K}(\mathbf{G}_{\mathrm{ii}}^{L})\right)\right)}_{\text{Sparse GP-ELBO}} \\
&\quad \underbrace{- \sum_{\ell=1}^{L} \nu_{\ell}\, \mathrm{D}_{\mathrm{KL}}\left(\mathcal{N}\left(\mathbf{0}, \mathbf{G}_{\mathrm{ii}}^{\ell}\right)\|\mathcal{N}\left(\mathbf{0}, \mathbf{K}(\mathbf{G}_{\mathrm{ii}}^{\ell-1})\right)\right)}_{\text{DKM Regularisation Terms}}.
\end{aligned}
\tag{67}
$$

We can see that with the full approximate posterior over the inducing outputs, the inducing objective can be neatly split up into the ELBO for a sparse shallow GP (the top layer) and a chain of regularisation terms for the hidden layers of the DKM.

In practice, we use the same $\mathbf{A}_{\lambda} = \mathbf{A}$ for all output features $\lambda$ to save memory and computational cost. We also use minibatching, so we normalise the objective by the total number of datapoints, meaning that computing our objective value on the minibatch provides an estimate for the fullbatch update.

Therefore, the mean minibatch objective we actually use in our implementation is

$$
\begin{aligned}
&\mathcal{L}_{\mathrm{ind}}^{\mathrm{batch}}(\mathbf{F}_{\mathrm{i}}^{L+1}, \mathbf{G}_{\mathrm{ii}}^{1}, \dots, \mathbf{G}_{\mathrm{ii}}^{L}) \\
&= \frac{1}{\text{Batch size}} \sum_{p \in \text{batch}} \mathbb{E}_{\mathrm{Q}}\left[\log \mathrm{P}\left(\mathbf{y}_{\mathrm{p}}|\mathbf{F}_{\mathrm{t}}^{L+1}\right)\right] \\
&\quad - \frac{1}{\text{Trainset Size}}\left[\sum_{\lambda=1}^{N_{L+1}} \mathrm{D}_{\mathrm{KL}}\left(\mathcal{N}\left(\boldsymbol{\mu}_{\lambda}, \mathbf{A}\right)\|\mathcal{N}\left(\mathbf{0}, \mathbf{K}(\mathbf{G}_{\mathrm{ii}}^{\ell-1})\right)\right) + \sum_{\ell=1}^{L} \nu_{\ell}\, \mathrm{D}_{\mathrm{KL}}\left(\mathcal{N}\left(\mathbf{0}, \mathbf{G}_{\mathrm{ii}}^{\ell}\right)\|\mathcal{N}\left(\mathbf{0}, \mathbf{K}(\mathbf{G}_{\mathrm{ii}}^{\ell-1})\right)\right)\right].
\end{aligned}
\tag{68}
$$

### D.2 LIKELIHOOD FUNCTIONS

We considered 2 different approaches to the likelihood: Gaussian (mirroring the approach in the early NNGP work, including Lee et al. 2017 and Matthews et al. 2018), and a Monte-Carlo estimate of a categorical likelihood, which is more natural for classification tasks.

Computing the GP-ELBO with these different likelihoods is not trivial, especially with inducing points. In this section, we detail precisely how these 2 different likelihood functions can be incorporated into the GP-ELBO, including showing that the GP-ELBO with a Gaussian likelihood has a closed form, and that the categorical likelihood can be approximated using Monte-Carlo estimation. Before we do this, recall that the inducing DKM objective in Eq. 67 contains the term

$$
\mathbb{E}_{\mathrm{Q}}\left[\log \mathrm{P}\left(\mathbf{Y}_{\mathrm{t}}|\mathbf{F}_{\mathrm{t}}^{L+1}\right)\right]
\tag{69}
$$

which is the expected log-likelihood under our variational joint distribution $Q = Q(\mathbf{F}_{\mathrm{t}}^{L+1}|\mathbf{F}_{\mathrm{i}}^{L+1}, \mathbf{G}^{L})Q(\mathbf{F}_{\mathrm{i}}^{L+1})$ over top-layer features $\mathbf{F}^{L+1}$, given the previous layer $\mathbf{G}^{L}$. Concretely, we choose $Q(\mathbf{F}_{\mathrm{i}}^{L+1})$ as in Eq. 66, and $Q(\mathbf{F}_{\mathrm{t}}^{L+1}|\mathbf{F}_{\mathrm{i}}^{L+1}, \mathbf{G}^{L}) = P(\mathbf{F}_{\mathrm{t}}^{L+1}|\mathbf{F}_{\mathrm{i}}^{L+1}, \mathbf{G}^{L})$, i.e. we use the same conditional Gaussian formula for test points given the inducing points as the true distribution.

We need only consider the predictive term (Eq. 69) since no other terms in the objective (Eq. 67) depend on our choice of likelihood function $P(\mathbf{Y}_{\mathrm{t}}|\mathbf{F}_{\mathrm{t}})$. With this in mind, we now derive the expected log-likelihood term for the two likelihood functions we consider.

### D.2.1 GAUSSIAN LIKELIHOOD

This is the same likelihood function used in the original DKM paper Yang et al. (2023), though they did not provide a detailed derivation of the closed-form expressions used to compute it. Since some readers will find this useful, we provide one here. In order to compute the DKM objective function, we must compute the predictive term

$$\mathbb{E}_{Q(\mathbf{F}_t^{L+1}|\mathbf{F}_i^{L+1},\mathbf{G}^L)Q(\mathbf{F}_i^{L+1})}\left[\log P(\mathbf{Y}_t|\mathbf{F}_t^{L+1})\right] \tag{70}$$

where, for a Gaussian likelihood,

$$P(\mathbf{Y}_t|\mathbf{F}_t^{L+1}) = \prod_{\lambda=1}^{N_{L+1}} \mathcal{N}(\mathbf{y}_\lambda; \mathbf{f}_\lambda^{t;L+1}, \sigma^2\mathbf{I}). \tag{71}$$

Substituting the definition of the multivariate normal log probability density and ignoring additive constants, we have

$$\mathbb{E}_{Q(\mathbf{F}_t^{L+1}|\mathbf{F}_i^{L+1},\mathbf{G}^L)Q(\mathbf{F}_i^{L+1})}\left[\log P(\mathbf{Y}_t|\mathbf{F}_t^{L+1})\right] \tag{72}$$

$$= -0.5 N_{L+1} P \log \sigma^2 - \frac{1}{2\sigma^2} \sum_{\lambda=1}^{N_{L+1}} \left\{ \mathbf{y}_\lambda^T \mathbf{y}_\lambda - 2\mathbf{y}_\lambda^T \mathbb{E}_Q[\mathbf{f}_\lambda] + \mathbb{E}_Q\left[\mathbf{f}_\lambda^T \mathbf{f}_\lambda\right] \right\} \tag{73}$$

where we use $\mathbb{E}_Q$ to mean $\mathbb{E}_{Q(\mathbf{F}_t^{L+1}|\mathbf{F}_i^{L+1},\mathbf{G}^L)Q(\mathbf{F}_i^{L+1})}$, and to avoid unruly notation we drop the layer and train/test labels from $\mathbf{f}_\lambda^{t;L+1}$ to just write $\mathbf{f}_\lambda$. This sum can be further decomposed into separate terms for each data point (which allows minibatching):

$$- 0.5 N_{L+1} P \log \sigma^2 - \frac{1}{2\sigma^2} \sum_{j=1}^{P} \sum_{\lambda=1}^{N_{L+1}} \left\{ y_{j\lambda}^2 - 2y_{t\lambda}\mathbb{E}_Q[f_{j\lambda}] + \mathbb{E}_Q\left[f_{j\lambda}^2\right] \right\} \tag{74}$$

$$= -0.5 N_{L+1} P \log \sigma^2 - \frac{1}{2\sigma^2} \sum_{j=1}^{P} \sum_{\lambda=1}^{N_{L+1}} \left\{ y_{j\lambda}^2 - 2y_{j\lambda}\mathbb{E}_Q[f_{j\lambda}] + \text{Var}_Q[f_{j\lambda}] + \mathbb{E}_Q[f_{j\lambda}]^2 \right\} \tag{75}$$

$$= -0.5 N_{L+1} P \log \sigma^2 - \frac{1}{2\sigma^2} \sum_{j=1}^{P} \sum_{\lambda=1}^{N_{L+1}} \left\{ (y_{j\lambda} - \mathbb{E}_Q[f_{j\lambda}])^2 + \text{Var}_Q[f_{j\lambda}] \right\}. \tag{76}$$

Hence to compute the predictive term we will need to know the mean and variance of $f_{j\lambda}$ (the test points $f_{j\lambda}^{t;L+1}$) under the joint distribution $Q$ (joint over both training and inducing). We assumed earlier that our inducing outputs $\mathbf{f}_\lambda^i$ have variational distribution

$$Q(\mathbf{f}_\lambda^i) = \mathcal{N}(\mathbf{f}_\lambda^i; \boldsymbol{\mu}_\lambda, \mathbf{A}_\lambda). \tag{77}$$

Since the conditional mean and covariance of $\mathbf{f}_\lambda^i$ given the inducing $\mathbf{f}_\lambda^i$ are just the conditional Gaussian formulae, this is very simple to compute via the laws of total expectation and total variance:

$$\begin{aligned}
\mathbb{E}_Q[\mathbf{f}_\lambda^t] &= \mathbb{E}\left[\mathbb{E}[\mathbf{f}_\lambda^t|\mathbf{f}_\lambda^i]\right] \\
&= \mathbb{E}[\mathbf{K}_{ti}\mathbf{K}_{ii}^{-1}\mathbf{f}_\lambda^i] \\
&= \mathbf{K}_{ti}\mathbf{K}_{ii}^{-1}\boldsymbol{\mu}_\lambda \\
\text{Var}[\mathbf{f}_\lambda^t] &= \mathbb{E}\left[\text{Var}[\mathbf{f}_\lambda^t|\mathbf{f}_\lambda^i]\right] + \text{Var}\left[\mathbb{E}[\mathbf{f}_\lambda^t|\mathbf{f}_\lambda^i]\right] \\
&= \mathbf{K}_{tt} - \mathbf{K}_{ti}\mathbf{K}_{ii}^{-1}\mathbf{K}_{it} + \text{Var}\left[\mathbf{K}_{ti}\mathbf{K}_{ii}^{-1}\mathbf{f}_\lambda^i\right] \\
&= \mathbf{K}_{tt} - \mathbf{K}_{ti}\mathbf{K}_{ii}^{-1}\mathbf{K}_{it} + \mathbf{K}_{ti}\mathbf{K}_{ii}^{-1}\text{Var}\left[\mathbf{f}_\lambda^i\right]\mathbf{K}_{ii}^{-1}\mathbf{K}_{it} \\
&= \mathbf{K}_{tt} - \mathbf{K}_{ti}\mathbf{K}_{ii}^{-1}\mathbf{K}_{it} + \mathbf{K}_{ti}\mathbf{K}_{ii}^{-1}\mathbf{A}_\lambda\mathbf{K}_{ii}^{-1}\mathbf{K}_{it}
\end{aligned} \tag{78, 79}$$

Combining this with Eq. 76 gives us our concrete formula. Alternatively, one can use these quantities to generate samples of $\mathbf{f}_\lambda^{t;L+1}$, which is multivariate Gaussian, and then compute the Gaussian likelihood via Monte-Carlo estimation. We used the latter strategy in our experiments using Gaussian likelihoods, since we are forced to sample for the categorical likelihood anyway.

### D.2.2 Categorical likelihood

We want to use a categorical distribution as our likelihood function, where the probabilities for each class are given by the softmax of our final layer features:

$$\mathrm{P}\left(\mathbf{Y}_t | \mathbf{F}_t^{L+1}\right) = \prod_{j=1}^{P} \sigma_{y_j}(\mathbf{f}_{j:}^{t;L+1}) \tag{80}$$

$$= \prod_{j=1}^{P} \frac{\exp f_{j,y_j}^{t;L+1}}{\sum_{\lambda=1}^{N_{L+1}} \exp f_{j,\lambda}^{t;L+1}} \tag{81}$$

Hence the expected log-likelihood would be

$$\mathbb{E}_Q\left[\log \mathrm{P}\left(\mathbf{Y}|\mathbf{F}_t^{L+1}\right)\right] = \sum_{j=1}^{P}\left\{\mathbb{E}_Q\left[f_{j,y_j}^{t;L+1}\right] - \mathbb{E}_Q\left[\log \sum_{\lambda=1}^{N_{L+1}} \exp f_{j,\lambda}^{t;L+1}\right]\right\} \tag{82}$$

The log-exp term is problematic, since we cannot compute its expectation in closed-form. However, since $\mathbf{f}_\lambda^{t;L+1}$ is just multivariate Gaussian distributed, with mean and covariance given by Eq. 78 and Eq. 79 respectively, we can easily generate samples and just compute a Monte-Carlo estimate of Eq. 82.

## E Flattening the spatial dimension at the final layer

### E.1 Linear

As in neural networks, we must collapse the spatial structure at the final layer, converting a $S_L P_t \times S_L P_t$ matrix to $P_t \times P_t$, so that we may perform the final classification. Convolution with zero-padding, and $D = S_L$ is equivalent to a linear top-layer. It returns a feature map with only a single spatial location. Hence we obtain similar expressions to our usual convolutional kernel. The tt'th part of these expressions was known to (Novak et al., 2018; Garriga-Alonso et al., 2018), but the inducing components (specifically $\mathbf{\Lambda}_{ii}$ and $\mathbf{\Lambda}_{it}$) are specific to our scheme.

$$\Xi_{i,j}^{ii}(\mathbf{G}^L) = \mathbb{E}\left[F_{i,\lambda}^{i;L+1} F_{j,\lambda}^{i;L+1} | \mathbf{H}^L\right] = \frac{1}{S_L} \sum_{s \in \mathcal{D}} \sum_{i'=1}^{P_i^L} \sum_{j'=1}^{P_i^L} C_{si,i'}^{L+1} C_{sj,j'}^{L+1} G_{i'j'}^{ii;L}. \tag{83}$$

$$\Xi_{i,j}^{it}(\mathbf{G}^L) = \mathbb{E}\left[F_{i,\lambda}^{i;L+1} F_{j,\lambda}^{t;L+1} | \mathbf{H}^L\right] = \frac{1}{S_L} \sum_{s} \sum_{i'=1}^{P_i^L} C_{si,i'}^{L+1} G_{i'j s}^{it;L} \tag{84}$$

$$\Xi_{i,j}^{tt}(\mathbf{G}^L) = \mathbb{E}\left[F_{i,\lambda}^{t;L+1} F_{j,\lambda}^{t;L+1} | \mathbf{H}^L\right] = \frac{1}{S_L} \sum_{s} G_{is,js}^{tt;L} \tag{85}$$

### E.2 Only diagonals of test/train blocks are needed

Naively applying these equations is very expensive, as we still need to work with $P_t S_\ell \times P_t S_\ell$ covariance matrices in the tt'th block. The key trick is that we only need to represent the diagonal of the tt'th matrices. In particular, note that Eq. (85) indicates that we only need the $P_t^2 S_L$ elements $G_{is,js}^{tt}$ corresponding to the covariance across images at the same spatial locations. Furthermore, the standard DKM derivations imply that (for IID likelihoods) we only need the diagonal of $\mathbf{G}_{tt}$, meaning that in this case we only need the $P_t S_L$ elements $G_{is,is}^{tt}$ corresponding to the variance of each image at each spatial location. This is only at the output layer, but as in infinite CNNs (Novak et al., 2018; Garriga-Alonso et al., 2018), it propagates backwards: if we only need $G_{is,is}^{tt;\ell}$ at one layer, then we only need $G_{is,is}^{tt;\ell-1}$ at the previous layer.

$$\Gamma_{is,is}(\mathbf{\Omega}_t) = \frac{1}{D} \sum_{d \in \mathcal{D}} \Omega_{i(s+d),i(s+d)}^t \tag{86}$$

### E.3 GLOBAL AVERAGE POOLING

It is common practice in convolutional neural network top-layers to use global average pooling, where we take the mean over the spatial dimension, and only learn weights on the channels (in contrast to a linear final layer where we would have a separate weight for each pixel). In this section we explain how this can be done for the convolutional DKM. Note that here we use $F_{i,r\mu}^L$ to denote the input to the global average pooling operation, which would usually be the result of a convolution layer, but the reader should focus on the operation of global average pooling rather the specific letters we choose to use here. For notational convenience, we let $S = S_L$ in the following derivations.

#### E.3.1 FULL-RANK

For features, we define global average pooling as

$$F_{i,\lambda}^{\text{GAP}} = \tfrac{1}{S} \sum_{r\mu} F_{i,r\mu}^L W_{\mu,\lambda}^{\text{GAP}}. \tag{87}$$

where, $\mathbf{W}^{\text{GAP}} \in \mathbb{R}^{N_L \times N_{L+1}}$ are IID Gaussian such that

$$E\left[W_{\mu,\lambda}^{\text{GAP}} W_{\nu,\lambda}^{\text{GAP}}\right] = \tfrac{1}{N_L} \delta_{\mu\nu}. \tag{88}$$

Now, computing the covariance $\Xi_{ij}$ (again, do not attribute too much meaning to the choice of letters), we have

$$\Xi_{ij} = \mathbb{E}\left[F_{i,\lambda}^{\text{GAP}} F_{j,\lambda}^{\text{GAP}}\right] \tag{89}$$

$$= \tfrac{1}{S^2} \mathbb{E}\left[\left(\sum_{r\mu} F_{i,r\mu}^L W_{\mu,\lambda}^{\text{GAP}}\right)\left(\sum_{s\nu} F_{j,s\nu}^L W_{\nu,\lambda}^{\text{GAP}}\right)\right]. \tag{90}$$

Rearranging the sums, we obtain

$$\Xi_{ij} = \tfrac{1}{S^2} \sum_{r\mu,s\nu} F_{i,r\mu}^L F_{j,s\nu}^L \mathbb{E}\left[W_{\mu,\lambda}^{\text{GAP}} W_{\nu,\lambda}^{\text{GAP}}\right] \tag{91}$$

which, by substituting the covariance of the weights (Eq. 88) becomes

$$\Xi_{ij} = \tfrac{1}{S^2} \sum_{r\mu,s\nu} F_{i,r\mu}^L F_{j,s\nu}^L \tfrac{1}{N_L} \delta_{\mu\nu}. \tag{92}$$

Finally, combining the delta-function and sum, we obtain

$$\Xi_{ij} = \tfrac{1}{S^2 N_L} \sum_{rs,\mu} F_{i,r\mu}^L F_{j,s\mu}^L \tag{93}$$

$$= \tfrac{1}{S^2} \sum_{rs} G_{ir,js}^L \tag{94}$$

i.e. we simply take the mean over both spatial indices (remember there are two because our covariance is comparing pixels to pixels).

#### E.3.2 INDUCING

Since global average pooling eliminates the spatial dimensions (or equivalently shares the same weights across all spatial locations within the same image), the inducing points can be directly multiplied by the weights, and we do not need any $\mathbf{C}$ weights to "map" them up to a spatial structure:

$$F_{i,\lambda}^{\text{i;GAP}} = \sum_{\mu} W_{\mu,\lambda}^{\text{GAP}} F_{i,\mu}^{\text{i;}L}. \tag{95}$$

The inducing-inducing block $\boldsymbol{\Lambda}_{\text{ii}}$ is straightforward:

$$\Xi_{ij}^{\text{ii}} = \mathbb{E}\left[F_{i,\lambda}^{\text{i;GAP}} F_{j,\lambda}^{\text{i;GAP}}\right] \tag{96}$$

$$= \mathbb{E}\left[\sum_\mu W_{\mu,\lambda}^{\text{GAP}} F_{i,\mu}^{\text{i};L} \sum_\nu W_{\nu,\lambda}^{\text{GAP}} F_{i,\nu}^{\text{i};L}\right] \tag{97}$$

$$= \sum_{\mu\nu} F_{i,\mu}^{\text{i};L} F_{i,\nu}^{\text{i};L} \, \mathbb{E}\left[W_{\mu,\lambda}^{\text{GAP}} W_{\nu,\lambda}^{\text{GAP}}\right] \tag{98}$$

$$= \sum_{\mu\nu} F_{i,\mu}^{\text{i};L} F_{i,\nu}^{\text{i};L} \tfrac{1}{N_L} \delta_{\mu\nu} \tag{99}$$

$$= G_{ij}^{\text{ii};L}, \tag{100}$$

i.e. it is just the same as the input. Since our inducing points have no spatial structure, it makes sense that global average pooling would do nothing to them.

The inducing-test/train block $\boldsymbol{\Lambda}_{\text{it}}$ just averages over the single spatial index:

$$\Xi_{i,j}^{\text{it}} = \mathbb{E}\left[F_{i,\lambda}^{\text{i;GAP}} F_{j,\lambda}^{\text{t;GAP}}\right] \tag{101}$$

$$= \mathbb{E}\left[\sum_\mu W_{\mu,\lambda}^{\text{GAP}} F_{i,\mu}^{\text{i};L} \tfrac{1}{S} \sum_{s\nu} F_{i,s\nu}^{\text{t};L} W_{\nu,\lambda}^{\text{GAP}}\right] \tag{102}$$

$$= \tfrac{1}{S} \mathbb{E}\left[\sum_{\mu\nu s} W_{\mu,\lambda}^{\text{GAP}} W_{\nu,\lambda}^{\text{GAP}} F_{i,\mu}^{\text{i};L} F_{i,s\nu}^{\text{t};L}\right] \tag{103}$$

$$= \tfrac{1}{SN_L} \mathbb{E}\left[\sum_{\mu s} F_{i,\mu}^{\text{i};L} F_{i,s\mu}^{\text{t};L}\right] \tag{104}$$

$$= \tfrac{1}{S} \sum_s G_{i,js}^{\text{it};L} \tag{105}$$

And the diagonal of $\boldsymbol{\Lambda}_{\text{tt}}$ (recall we need only store the diagonal for IID likelihoods) averages over both inputted spatial indices:

$$\Xi_{i,i}^{\text{tt}} = \mathbb{E}\left[F_{i,\lambda}^{\text{t;GAP}} F_{i,\lambda}^{\text{t;GAP}}\right] \tag{106}$$

$$= \tfrac{1}{S^2} \mathbb{E}\left[\sum_{s\mu} F_{i,s\mu}^{\text{t};L} W_{\mu,\lambda}^{\text{GAP}} \sum_{r\nu} F_{i,r\nu}^{\text{t};L} W_{\nu,\lambda}^{\text{GAP}}\right] \tag{107}$$

$$= \tfrac{1}{S^2} \mathbb{E}\left[\sum_{sr\mu\nu} W_{\mu,\lambda}^{\text{GAP}} W_{\nu,\lambda}^{\text{GAP}} F_{i,s\mu}^{\text{t};L} F_{i,r\nu}^{\text{t};L}\right] \tag{108}$$

$$= \tfrac{1}{S^2 N_\ell} \mathbb{E}\left[\sum_{sr\mu} F_{i,s\mu}^{\text{t};L} F_{i,r\mu}^{\text{t};L}\right] \tag{109}$$

$$= \tfrac{1}{S^2} \sum_{sr} G_{is,ir}^{\text{tt};L} \tag{110}$$

However, there is a problem. This does not depend only on the $P_{\text{t}} \times S$ diagonal of the input, but also on the covariance of all pixels within a test image, which means we actually require an object of size $P_{\text{t}} \times S \times S$ versus the full matrix $\mathbf{G}_{\text{tt}}$ of size $P_{\text{t}} \times P_{\text{t}} \times S \times S$ which is still bad since $S$ is usually much larger than either $P_{\text{t}}$ or $P_{\text{i}}$.

To solve this problem, we use a corrected Nyström approximation on the input:

$$\mathbf{G}_{\text{tt}} \approx \mathbf{G}_{\text{it}}^T \mathbf{G}_{\text{ii}}^{-1} \mathbf{G}_{\text{it}} + \mathbf{D}. \tag{111}$$

where $\mathbf{D}$ is a diagonal matrix that corrects the diagonal to match the true diagonal of $\mathbf{G}_{\text{tt}}$, which we know.

However, this still seems like somewhat of a problem, since the Nyström approximation of $\mathbf{G}_{\text{tt}}$ is still of size $P_{\text{i}} \times P_{\text{t}} \times S \times S$. Luckily, it turns out that, since we only need the diagonal of $\mathbf{K}_{\text{tt}}$, we don't need to compute the entire Nyström approximation:

$$\Xi_{i,j}^{\text{tt}} = \tfrac{1}{S^2} \sum_{sr} G_{is,jr}^{\text{tt};L} \tag{112}$$

Apply the Nyström approximation:

$$\Xi_{i,j}^{\text{tt}} \approx \tfrac{1}{S^2} \sum_{sr} \left[ \sum_{i'j'} \left( G_{i',is}^{\text{it};L} \left[ \mathbf{G}_{\text{ii}}^L \right]_{i',j'}^{-1} G_{j,j'r}^{\text{it};L} \right) + D_{is,is} \delta_{ij} \delta_{sr} \right] \tag{113}$$

$$= \sum_{i'j'} \left( \tfrac{1}{S} \sum_s G_{i',is}^{\text{it};L} \right) \left[ \mathbf{G}_{\text{ii}}^L \right]_{i',j'}^{-1} \left( \tfrac{1}{S} \sum_r G_{j,j'r}^{\text{it};L} \right) + \tfrac{1}{S^2} \sum_s D_{is,is} \delta_{ij} \tag{114}$$

$$\tag{115}$$

$$= \sum_{i'j'} \Xi_{i',i}^{\text{it}} \left[ \mathbf{G}_{\text{ii}}^L \right]_{i',j'}^{-1} \Xi_{j,j'}^{\text{it}} + \tfrac{1}{S^2} \sum_s D_{is,is} \delta_{ij}. \tag{116}$$

And because we only need the diagonal,

$$\Xi_{i,i}^{\text{tt}} = \sum_{i'j'} \Xi_{i',i}^{\text{it}} \left[ \mathbf{G}_{\text{ii}}^L \right]_{i',j'}^{-1} \Xi_{i,j'}^{\text{it}} + \tfrac{1}{S^2} \sum_s D_{is,is}. \tag{117}$$

With that problem solved, we can now practically apply global average pooling to our Gram matrices at the top-layer.

# F    EXPERIMENTAL RESULTS - FURTHER METRICS

In this appendix we give further metrics (test log-likelihood, train accuracy, and train log-likelihood) for the experiments in the main text. All experiments were run for 4 different random seeds, and we report the means with standard errors. See Section 5 for further details of these experiments.

As in the main text, for all tables and for each dataset, we identify the best configuration, and then bold all those configurations whose performance was statistically similar to the best, using a one-tailed Welch's t-test with confidence level 5%.

## F.1    REGULARISATION STRENGTH

Test accuracy for regularisation strength were already given in the main text (Table 1). We give the remaining metrics in Table 4.

As we can see from the performance metrics, test log-likelihoods are maximised by using $\nu = 1$, though this comes at a slight performance hit to test accuracy (Table 1). For smaller values, higher train performance over test performance suggests overfitting is occurring (though this seems to only negatively affect the log-likelihoods and not the accuracies). It may be an interesting future direction to investigate alternative formulations for the regularisation term to see if this can be improved upon.

## F.2    BATCH NORMALISATION TYPE

The test metrics for normalisation and rescaling were already given in the main text (Table 1). We give the remaining metrics for normalisation in Table 5 and for rescaling in Table 6.

We chose the combinations of variants by requiring that the inducing blocks and test-train blocks used the same type of normalisation and scaling. In cases where we are using image specific variants of normalisation on the test-train blocks that don't exist for the inducing blocks, we use the following mapping: "image" normalisation in the test-train block corresponds to "local" normalisation in the inducing block, since it doesn't average across the datapoints in the batch, whilst "location" normalisation corresponds to "Batch" normalisation in the inducing block since it does average across

Table 4: Train accuracies (%) and log-likelihoods using different regularisation strengths.

| Metric | $\nu$ | MNIST | CIFAR-10 | CIFAR-100 |
|---|---|---|---|---|
| Test LL | $\infty$ | -0.9026 ± 0.0039 | -1.3546 ± 0.0055 | -2.8909 ± 0.0105 |
| | $10^4$ | -0.6393 ± 0.0158 | -1.3830 ± 0.0063 | -2.9914 ± 0.0030 |
| | $10^3$ | -0.2488 ± 0.0014 | -1.1972 ± 0.0018 | -2.6351 ± 0.0206 |
| | $10^2$ | -0.0942 ± 0.0029 | -0.8322 ± 0.0003 | -2.0077 ± 0.0058 |
| | $10^1$ | -0.0380 ± 0.0013 | -0.5070 ± 0.0058 | -1.5307 ± 0.0133 |
| | $10^0$ | **-0.0263 ± 0.0006** | **-0.3885 ± 0.0022** | **-1.4638 ± 0.0093** |
| | $10^{-1}$ | -0.0294 ± 0.0011 | -0.4652 ± 0.0051 | -1.7877 ± 0.0309 |
| | $10^{-2}$ | -0.0344 ± 0.0013 | -0.5048 ± 0.0102 | -1.9923 ± 0.0310 |
| | $10^{-3}$ | -0.0356 ± 0.0011 | -0.5295 ± 0.0049 | -2.0162 ± 0.0144 |
| | $10^{-4}$ | -0.0371 ± 0.0012 | -0.5279 ± 0.0017 | -2.0468 ± 0.0107 |
| | $0$ | -0.0359 ± 0.0010 | -0.5280 ± 0.0072 | -2.0552 ± 0.0122 |
| Train Acc. | $\infty$ | 73.10 ± 0.09 | 52.99 ± 0.22 | 30.19 ± 0.25 |
| | $10^4$ | 82.03 ± 0.55 | 51.30 ± 0.31 | 27.75 ± 0.13 |
| | $10^3$ | 92.76 ± 0.04 | 58.24 ± 0.06 | 34.72 ± 0.41 |
| | $10^2$ | 96.82 ± 0.09 | 70.66 ± 0.07 | 50.09 ± 0.18 |
| | $10^1$ | 98.78 ± 0.03 | 83.84 ± 0.23 | 65.20 ± 0.57 |
| | $10^0$ | 99.48 ± 0.02 | 93.22 ± 0.02 | 82.48 ± 0.51 |
| | $10^{-1}$ | 99.77 ± 0.00 | 97.45 ± 0.07 | 91.07 ± 0.34 |
| | $10^{-2}$ | 99.87 ± 0.01 | 98.38 ± 0.08 | **93.72 ± 0.37** |
| | $10^{-3}$ | **99.90 ± 0.01** | **98.65 ± 0.04** | **94.18 ± 0.28** |
| | $10^{-4}$ | **99.90 ± 0.01** | **98.67 ± 0.06** | **94.34 ± 0.26** |
| | $0$ | **99.91 ± 0.00** | **98.70 ± 0.05** | **94.41 ± 0.25** |
| Train LL | $\infty$ | -0.8735 ± 0.0021 | -1.3939 ± 0.0036 | -2.9417 ± 0.0073 |
| | $10^4$ | -0.6149 ± 0.0169 | -1.4230 ± 0.0067 | -3.0430 ± 0.0032 |
| | $10^3$ | -0.2589 ± 0.0006 | -1.2351 ± 0.0012 | -2.6898 ± 0.0188 |
| | $10^2$ | -0.1084 ± 0.0027 | -0.8547 ± 0.0003 | -1.9135 ± 0.0096 |
| | $10^1$ | -0.0382 ± 0.0008 | -0.4651 ± 0.0053 | -1.2463 ± 0.0212 |
| | $10^0$ | -0.0164 ± 0.0005 | -0.1944 ± 0.0012 | -0.5835 ± 0.0171 |
| | $10^{-1}$ | -0.0072 ± 0.0001 | -0.0729 ± 0.0018 | -0.2833 ± 0.0102 |
| | $10^{-2}$ | -0.0041 ± 0.0001 | -0.0462 ± 0.0019 | **-0.1980 ± 0.0100** |
| | $10^{-3}$ | -0.0033 ± 0.0000 | **-0.0387 ± 0.0012** | **-0.1814 ± 0.0083** |
| | $10^{-4}$ | -0.0032 ± 0.0001 | **-0.0377 ± 0.0013** | **-0.1758 ± 0.0081** |
| | $0$ | **-0.0029 ± 0.0001** | **-0.0365 ± 0.0011** | **-0.1736 ± 0.0079** |

the batch. For rescaling, the are no longer "local" or "image" variants for the test-train block, since it does not make sense to have a scalar for individual datapoints in a batch that will persist across arbitrary batches. Hence, when we are using "local" on the inducing block, we test all possible variants for the test-train block.

Our batchnorm schemes were originally introduced to provide smoother training dynamics, but the story the results tell is mixed. Sometimes using no normalisation or scaling results in worse training performance, like on CIFAR-100, so we decided that it seemed sensible to use the simplest "Batch" variants for the final experiments (Table 2 in the main text, and Table 9 for other metrics), which seemed to perform well across all the datasets.

### F.3 FINAL LAYER AFTER CONVOLUTIONS

Test accuracies for the final layer type were given in the main text (Table 1). We give the rest of the metrics in Table 7.

Our global average pooling layer requires the use of a Nyström approximation. To ensure this was not adversely affecting performance, we also test a final layer that performs a convolution with kernel size equal to the feature map size (analogous to flattening and using a linear layer in a CNN) which

Table 5: Train accuracies (%) and log-likelihoods using different types of normalisation (Inducing / Train-Test).

| Metric | Ind. / Train-Test | MNIST | CIFAR-10 | CIFAR-100 |
|---|---|---|---|---|
| Test LL | Batch / Batch | **-0.0263 ± 0.0006** | **-0.3885 ± 0.0022** | -1.4638 ± 0.0093 |
| | Batch / Location | **-0.0284 ± 0.0009** | **-0.3850 ± 0.0027** | -1.4708 ± 0.0129 |
| | Local / Image | **-0.0277 ± 0.0008** | **-0.3863 ± 0.0011** | -1.4868 ± 0.0195 |
| | Local / Local | **-0.0270 ± 0.0009** | -0.3952 ± 0.0034 | -1.5365 ± 0.0118 |
| | None / None | **-0.0276 ± 0.0008** | -0.4025 ± 0.0046 | **-1.4316 ± 0.0007** |
| Train Acc. | Batch / Batch | **99.48 ± 0.02** | **93.22 ± 0.02** | 82.48 ± 0.51 |
| | Batch / Location | 99.44 ± 0.01 | **92.94 ± 0.17** | 82.42 ± 0.61 |
| | Local / Image | **99.51 ± 0.01** | **93.06 ± 0.17** | **83.88 ± 0.27** |
| | Local / Local | 99.41 ± 0.01 | **92.99 ± 0.17** | **83.69 ± 0.20** |
| | None / None | **99.49 ± 0.01** | 92.48 ± 0.13 | 79.68 ± 0.16 |
| Train LL | Batch / Batch | **-0.0164 ± 0.0005** | **-0.1944 ± 0.0012** | -0.5835 ± 0.0171 |
| | Batch / Location | -0.0175 ± 0.0001 | **-0.2037 ± 0.0047** | -0.5855 ± 0.0205 |
| | Local / Image | **-0.0158 ± 0.0004** | **-0.1998 ± 0.0050** | **-0.5302 ± 0.0088** |
| | Local / Local | -0.0185 ± 0.0002 | **-0.1995 ± 0.0054** | **-0.5395 ± 0.0071** |
| | None / None | **-0.0157 ± 0.0002** | -0.2162 ± 0.0036 | -0.6821 ± 0.0067 |

Table 6: Train accuracies (%) and log-likelihoods using different types of rescaling after normalisation.

| Metric | Ind. / Train-Test | MNIST | CIFAR-10 | CIFAR-100 |
|---|---|---|---|---|
| Test LL | Batch / Batch | **-0.0263 ± 0.0006** | **-0.3885 ± 0.0022** | -1.4638 ± 0.0093 |
| | Batch / Location | **-0.0276 ± 0.0008** | **-0.3892 ± 0.0045** | -1.5375 ± 0.0149 |
| | Local / Batch | **-0.0257 ± 0.0005** | **-0.3930 ± 0.0025** | -1.4708 ± 0.0126 |
| | Local / Location | -0.0287 ± 0.0007 | **-0.3894 ± 0.0030** | -1.5050 ± 0.0176 |
| | Local / None | **-0.0261 ± 0.0010** | **-0.3894 ± 0.0023** | -1.4298 ± 0.0076 |
| | None / None | **-0.0275 ± 0.0008** | **-0.3904 ± 0.0039** | **-1.3514 ± 0.0017** |
| Train Acc. | Batch / Batch | **99.48 ± 0.02** | **93.22 ± 0.02** | **82.48 ± 0.51** |
| | Batch / Location | 99.43 ± 0.01 | **93.43 ± 0.09** | **81.69 ± 0.68** |
| | Local / Batch | **99.48 ± 0.01** | 92.63 ± 0.12 | **81.90 ± 0.15** |
| | Local / Location | 99.41 ± 0.02 | 92.82 ± 0.24 | **81.83 ± 0.24** |
| | Local / None | 99.44 ± 0.01 | **93.26 ± 0.08** | **81.31 ± 0.59** |
| | None / None | **99.46 ± 0.02** | 93.07 ± 0.07 | 78.99 ± 0.34 |
| Train LL | Batch / Batch | **-0.0164 ± 0.0005** | **-0.1944 ± 0.0012** | **-0.5835 ± 0.0171** |
| | Batch / Location | -0.0179 ± 0.0002 | **-0.1874 ± 0.0035** | **-0.6116 ± 0.0246** |
| | Local / Batch | **-0.0163 ± 0.0005** | -0.2099 ± 0.0036 | **-0.5993 ± 0.0051** |
| | Local / Location | -0.0185 ± 0.0003 | -0.2042 ± 0.0067 | **-0.6101 ± 0.0080** |
| | Local / None | -0.0178 ± 0.0005 | **-0.1950 ± 0.0011** | **-0.6251 ± 0.0212** |
| | None / None | -0.0174 ± 0.0002 | **-0.1953 ± 0.0027** | -0.7135 ± 0.0135 |

does not require any extra approximations. We found that global average pooling either matched or outperformed this method across the board, in line with common deep learning practice, despite the use of the Nyström approximation.

We also experimented with learning weights to "mix up" the inducing points as in our convolutional layers, but results were mixed, and it sometimes even made the model worse (presumably due to the increased complexity introduced to the optimisation problem by these extra weights).

Table 7: Train accuracies (%) and log-likelihoods using different final layers.

| Metric | Final Layer Type | MNIST | CIFAR-10 | CIFAR-100 |
|---|---|---|---|---|
| Test LL | Global Average Pooling | **-0.0263 ± 0.0006** | -0.3885 ± 0.0022 | -1.4638 ± 0.0093 |
| | Linear | **-0.0275 ± 0.0006** | **-0.3751 ± 0.0040** | -1.4053 ± 0.0087 |
| | GAP + Mixup | **-0.0275 ± 0.0004** | **-0.3836 ± 0.0008** | -1.4861 ± 0.0107 |
| | Linear + Mixup | **-0.0280 ± 0.0007** | -0.3875 ± 0.0032 | **-1.3734 ± 0.0065** |
| Train Acc | Global Average Pooling | **99.48 ± 0.02** | **93.22 ± 0.02** | **82.48 ± 0.51** |
| | Linear | **99.46 ± 0.02** | 92.39 ± 0.11 | 79.93 ± 0.30 |
| | GAP + Mixup | **99.49 ± 0.01** | **92.77 ± 0.23** | **83.41 ± 0.19** |
| | Linear + Mixup | **99.47 ± 0.01** | 92.07 ± 0.35 | 74.26 ± 0.21 |
| Train LL | Global Average Pooling | **-0.0164 ± 0.0005** | **-0.1944 ± 0.0012** | **-0.5835 ± 0.0171** |
| | Linear | **-0.0173 ± 0.0007** | -0.2197 ± 0.0035 | -0.6682 ± 0.0111 |
| | GAP + Mixup | **-0.0160 ± 0.0003** | **-0.2075 ± 0.0072** | **-0.5533 ± 0.0063** |
| | Linear + Mixup | -0.0169 ± 0.0004 | -0.2276 ± 0.0108 | -0.8785 ± 0.0048 |

Table 8: Train accuracies (%) for different likelihood functions. (log-likelihoods are not comparable). Bold shows values statistically similar to the maximum.

| Metric | Likelihood | MNIST | CIFAR-10 | CIFAR-100 |
|---|---|---|---|---|
| Train Acc. | Gaussian | 99.12 ± 0.01 | 88.85 ± 0.59 | 27.17 ± 0.67 |
| | Categorical | **99.48 ± 0.02** | **93.22 ± 0.02** | **82.48 ± 0.51** |

### F.4 LIKELIHOOD FUNCTION

Test accuracies for the choice of likelihood function were given in the main text (Table 1). We give the train accuracies in Table 8. Since we are modifying the likelihood function, it does not make sense to compare the log-likelihoods between these methods.

We tested two different likelihood functions: the Gaussian likelihood and the categorical likelihood with softmax probabilities. We estimated both the categorical likelihood and Gaussian likelihood via Monte-Carlo, though the Gaussian likelihood does permit a closed-form objective function. Details of both likelihood functions can be found in Appendix D.2. For the Monte-Carlo approximation, we used 1000 samples from the top layer features.

As expected for classification, the categorical likelihoods performed better than the Gaussian likelihood. In the case of CIFAR-100, which has 100 classes compared to CIFAR-10 and MNIST which have 10 classes, the Gaussian likelihood resulted in catastrophically bad training performance, which then carried over into test performance.

### F.5 FINAL EXPERIMENTS - NUMBER OF INDUCING POINTS

For the final experiments, using hyperparameters selected using the results from the previous experiments, test accuracies for different numbers of inducing points were given in the main text (Table 2). We give the other metrics in Table 9. Interestingly, whilst we observe good test accuracy on the largest models, the test log-likelihood actually degrades past a certain size of model. Since DKMs are not considered truly Bayesian due to the model mismatch introduced by the modified likelihood layer (Yang et al., 2023), we hypothesise that this is the model overfitting on the train log-likelihood and causing poor calibration.

Table 9: Train accuracies (%) and log-likelihoods using different numbers of inducing points.

| Metric | Ind. Points | MNIST | CIFAR-10 | CIFAR-100 |
|---|---|---|---|---|
| Test LL | 16 / 32 / 64 | **-0.0312 ± 0.0008** | -0.4960 ± 0.0047 | -1.7118 ± 0.0073 |
| | 32 / 64 / 128 | **-0.0365 ± 0.0025** | **-0.4548 ± 0.0049** | **-1.5182 ± 0.0085** |
| | 64 / 128 / 256 | -0.0443 ± 0.0007 | -0.6446 ± 0.0146 | -2.3413 ± 0.0656 |
| | 128 / 256 / 512 | -0.0603 ± 0.0019 | -0.7593 ± 0.0214 | -3.1398 ± 0.0491 |
| | 256 / 512 / 1024 | -0.0659 ± 0.0016 | -0.6939 ± 0.0154 | -2.4611 ± 0.0196 |
| | 512 / 1024 / 2048 | -0.0676 ± 0.0036 | -0.6502 ± 0.0125 | -2.0553 ± 0.0207 |
| Train Acc. | 16 / 32 / 64 | 99.45 ± 0.02 | 86.27 ± 0.21 | 58.22 ± 0.27 |
| | 32 / 64 / 128 | 99.76 ± 0.00 | 93.75 ± 0.04 | 75.15 ± 0.41 |
| | 64 / 128 / 256 | 99.94 ± 0.00 | 98.83 ± 0.11 | 94.76 ± 0.38 |
| | 128 / 256 / 512 | 99.99 ± 0.00 | 99.90 ± 0.01 | 99.83 ± 0.02 |
| | 256 / 512 / 1024 | **100.00 ± 0.00** | 99.98 ± 0.00 | **99.98 ± 0.00** |
| | 512 / 1024 / 2048 | **100.00 ± 0.00** | **99.99 ± 0.00** | **99.98 ± 0.00** |
| Train LL | 16 / 32 / 64 | -0.0166 ± 0.0005 | -0.3907 ± 0.0061 | -1.4938 ± 0.0098 |
| | 32 / 64 / 128 | -0.0075 ± 0.0003 | -0.1743 ± 0.0012 | -0.8324 ± 0.0144 |
| | 64 / 128 / 256 | -0.0020 ± 0.0001 | -0.0328 ± 0.0028 | -0.1631 ± 0.0114 |
| | 128 / 256 / 512 | -0.0004 ± 0.0000 | -0.0033 ± 0.0002 | -0.0104 ± 0.0007 |
| | 256 / 512 / 1024 | **-0.0002 ± 0.0000** | -0.0010 ± 0.0000 | -0.0026 ± 0.0001 |
| | 512 / 1024 / 2048 | **-0.0001 ± 0.0000** | **-0.0005 ± 0.0001** | **-0.0015 ± 0.0000** |

