# OpenReview forum: "Convolutional Deep Kernel Machines"
_ICLR.cc/2024/Conference — ICLR 2024 poster_

### Official Review · Reviewer_jw8S · 2023-10-25

**Soundness:** 3 good
**Presentation:** 4 excellent
**Contribution:** 3 good
**Rating:** 8
**Confidence:** 3

**Summary:**

The paper proposes convolutional deep kernel machines (DKMs) which builds upon a previous work of DKM while considers only settings with small-scale and low-dimensional data. The authors first introduce how DKM parameterizes the intermediate kernel matrices in NNGP limit by modifying the objective function. The convolutional DKM is then constructed by incorporating convolutional structure into the kernel function with Gaussian prior on the filter weights. To reduce its computational complexity, the authors leverage the idea of inducing point approach by eliminating spatial structure in inducing points while keeping spatial structure in the remaining data points. The proposed model is tested on several benchmark computer vision datasets with different sets of structures and hyperparameters and compared to other kernel-based models.

**Strengths:**

•	To my knowledge, the proposed inter-domain inducing point approach seems to be novel and work well in the given context.

•	The authors conduct a comprehensive study of different architectures and hyperparameters of the proposed method.

•	The paper is clearly written and well organized.

**Weaknesses:**

The scope and applications of convolutional DKM are somewhat limited. From my perspective, a main purpose of kernel-based methods (i.e., NNGP, NTK, DMK, etc.) is to help explain the learning mechanism of Bayesian NNs instead of giving state-of-the-art performance on regression or classification tasks (as can be observed in Table 3). It is thus desirable to briefly discuss how the proposed method contributes to this domain (i.e., explaining the learning scheme of Bayesian convolutional NNs).

**Questions:**

•	The authors state that DKMs are derived from deep GPs where the width of the intermediate layers goes to infinity, but it seems that the kernel function in Section 4 is calculated based on finite number of channels (i.e., $N_l$ is finite). Can authors provide some clarification?

•	It is not clear whether the kernel matrix $\textbf{K}\_{\text{Conv}}$ is positive-definite by defining $\textbf{F}_{i}^{l}$ using learnable parameter $\textbf{C}^{l}$, or at least a priori (i.e., before training). Can authors give some intuition on this?

•	Why is the convolutional DKM compared with other models only on CIFAR-10 but not CIFAR-100 in Table 3?

---

> ### Author Response · Authors · 2023-11-21
> **Author Response**
>
> Please see **Overall summary** at the top, which describes major changes, including dramatically improved performance on CIFAR-10 (from 91.7\% in the original manuscript to 92.7\% in this version), and a revamped introduction section that more clearly explains the motivation of our work.
>
> **Theoretical contributions of the convolutional DKM.**
> Absolutely!  We have rewritten the introduction to highlight that there is a large body of work in the NNGP and NTK literature, attempting to push up the performance of these methods.  There is good theoretical reason for doing this.  In particular, we want to know whether infinite-width neural networks capture the "essence" of what makes neural networks work so well, or whether there is something missing.  Indeed, it does seem that NNGPs underperform relative to finite NNs, suggesting that NNGPs are indeed missing something fundamental. One hypothesis is that this missing piece is representation learning.  However, we can't really be sure that this is the case, given the numerous practical differences between NNGPs and finite NNs.  One way to confirm this hypothesis would be to develop an infinite-width NNGP-like approach that somehow includes representation learning.  We do exactly that, and find that we do indeed get improved performance over NNGPs (92.7\% vs the best NNGP result of 91.2\% from Adlam et al. 2023). Furthermore, the DKM framework provides a handle (the regularisation strength hyperparameter $\nu$) to smoothly interpolate between the NNGP kernel regime and the fully-representation-learning regime (see Table 1).
>
>
> Questions:
> > The authors state that DKMs are derived from deep GPs where the width of the intermediate layers goes to infinity, but it seems that the kernel function in Section 4 is calculated based on finite number of channels (i.e.,  is finite). Can authors provide some clarification?
>
> Here, we're taking inspiration from neural networks to derive an inter-domain inducing point scheme for the infinite width limit, so we necessarily intermix the kernel and feature-based viewpoint.  There's two tricks here as we move from the feature-based to the kernel-based viewpoint.
>
> First, we use expressions like Eq. 15, which are valid when $\mathbf{H}$ has a finite or infinite number of features.
>
> Second, in e.g.\ Eq. 19 and 20, we use a standard trick from the NNGP literature.
> In particular, the distribution over all features (indexed $\lambda$) is IID.
> Therefore, we consider a particular feature, (the $\lambda$th feature) in Eq. 19 and 20, but the derivation applies to all (potentially infinitely many) features.
>
>
> > It is not clear whether the kernel matrix  is positive-definite by defining
>  using learnable parameter , or at least a priori (i.e., before training). Can authors give some intuition on this?
>
> The kernel defined in (Eq. 17) is just the covariance of a random vector, so it has to be positive semi-definite.
> Specifically, Eq. 17 can be rewritten as,
>
> $\mathbf{K}^{\rm{Conv}} = \mathbb{E}[\mathbf{f}_{\lambda}  \mathbf{f}\_{\lambda}^T]$
>
> where, $\mathbf{f}^T = (\mathbf{f}^{\rm i}\_\lambda \mathbf{f}^{\rm t}_\lambda)^T$.
> We use $C$ to define $\mathbf{f}^{\rm_i}$, but the kernel would be positive semi-definite for any choice of $\mathbf{f}^{\rm_i}$.
>
> > Why is the convolutional DKM compared with other models only on CIFAR-10 but not CIFAR-100 in Table 3?
>
> Previous methods take so long to run that they tend to only give results on CIFAR-10 (but we're happy to include comparisons on CIFAR-100 if you know of any relevant ones).

---

> > ### Comment · Reviewer_jw8S · 2023-11-22
> >
> > Thank you for your clarifications on my questions. I definitely think the transition from the feature-based to the kernel-based viewpoint deserves further elaboration. Apart from that, my assessment of the novelty and contribution of this work to the Bayesian learning of Neural Networks remains unchanged, and I'm open to raising the score if the authors add these explanations to the updated manuscript.

---

> > > ### Author Response · Authors · 2023-11-22
> > >
> > > Thanks for your response. We have revised our manuscript to incorporate these changes, which hopefully make this subtle point a bit clearer. In particular:
> > >
> > > Under Eq. (9), after we first define a kernel as the covariance of features, we write "Note that this implies all features (indexed by $\lambda$) are IID, so we consider an arbitrary feature $\lambda$. This is valid even if $N_\ell \to \infty$."
> > >
> > > Under Eq. (15), after we define the Gram matrices in terms of features, we write "Note that while the features in Eq. (14) can only be realised for finite $N_\ell$, Eq. (15) is valid as $N_\ell \to \infty$."
> > >
> > > And finally, just before Eqs. (18) and (19), we write "(we again emphasise that all features are IID, so the covariance is defined for an arbitrary feature $\lambda$)"
> > >
> > > Please let us know if you'd recommend any further changes.

---

### Official Review · Reviewer_hjfZ · 2023-10-30

**Soundness:** 3 good
**Presentation:** 3 good
**Contribution:** 2 fair
**Rating:** 5
**Confidence:** 4

**Summary:**

This paper formulates the convlutional deep kernel machines based on the framework of a very recent work (Yang et al. 2023), which presents a modified  Neural Network Gaussian Process (NNGP) for retaining a better representation learning. In this work, the authors formulate the convolutional operations in deep kernel machines built upon the convolutional kernel, apply the NNGP framework from (Yang et al. 2023) with an ELBO objective attained, further utilize the sparse GP technique with specific consideration to the convolutional kernel case for yielding a tracble training scheme with the induced points, and conduct experiments mainly mirroring ResNet20 with good results compared to other kernel techniques.

**Strengths:**

In general, the elements involved in the work and the used technique framework are clear. Despite of the "a bit giant" title, the content presented in the work reflect what the title conveys, so the focus of this work is  well presented.

**Weaknesses:**

However, for the work itself, I found a bit struggling to appreciate the (both theoretical and practical) benefits of applying the proposed method.

**Questions:**

### Major concerns:
1. In the presented content, it does not clearly elaborate or reveal the classical benefits of using kernel method (e.g., more interpretation, primal-dual representation and optimization, analytical results from RKHS, or [1] etc), nor provide practical values of using this method (e.g., varying the architecture besides ResNet20, a possibility of using a simple architecture).

Besides, Would it be possible  that the GP-relied framework in this deep kernel machine shall have advantages in dealing uncertainty in data/images with better robustness or generalization ability. I would suggest the authors to articulate this aspect.

[1] Bohn, B., Rieger, C., & Griebel, M. (2019). A representer theorem for deep kernel learning. The Journal of Machine Learning Research, 20(1), 2302-2333.

2. To my personal experience, the title of this paper looks too "giant", as there exist different lines of work which try to deepen kernel machines w.r.t. different principles, setups and goals. The DKM is basically relying on the NNGP framework (Yang et al. 2023) and the convolution kernel. Besides the cited work in the paper, parts of other work regarding deep kernel machines can involve (**not limited to**):

Mairal, Julien, et al. "Convolutional kernel networks. Advances in neural information processing systems 27 (2014),

Mairal, Julien. "End-to-end kernel learning with supervised convolutional kernel networks." Advances in neural information processing systems 29 (2016),

Chen, Dexiong, Laurent Jacob, and Julien Mairal. "Recurrent kernel networks." Advances in Neural Information Processing Systems 32 (2019), and some other works of Julien Mairal https://lear.inrialpes.fr/people/mairal/resources/pdf/Ohrid_CKN.pdf, where neural networks are involved quite a lot towards a deep architecture.

Suykens, Johan AK. "Deep restricted kernel machines using conjugate feature duality." Neural computation 29.8 (2017),

Tonin, Francesco, et al. "Deep Kernel Principal Component Analysis for Multi-level Feature Learning." arXiv preprint arXiv:2302.11220 (2023),

Achten, Sonny, et al. "Semi-Supervised Classification with Graph Convolutional Kernel Machines." arXiv preprint arXiv:2301.13764 (2023),

and other works of Johan AK Suykens, where the primal-dual framework retaining feature maps (e.g., NNs) in the primal and kernel tricks in the dual is a central idea. I would suggest to have a more comprehensive review on deep kernel machines in related work, though they are designing the model differently. It might also help to distinguish the unique benefits of the proposed method.


### Minor aspects:
1. A configuration figure presenting the convolutional kernel, the deep  architecture and the flow of optimization with sparse GP can be helpful for illustration I suppose.

2. Some typos and non-mathemcatical edits, e.g., the 2nd paragraph under Table 3 on page 8.

3. In the provided codes, there are different variants of nonlinear kernels. But I didn't notice a clear descriptions on how these nonlinear units are chosen or considered.

4. How about the empirical efficiency difference on varying the inducing points setups in Table 2?

5. In Table 1, many factors are investigated, and in different sections mostly the performances did not changge drastically in the compared techniques and were similar to other sections. The evaluation is comprehensive, but the conlusion is less focused, as we would hope to quickly get which factors are the most important ones. I these results don't change much and all attain good results, and then a mild suggestion or a global defaulting setting can be a take-away message.

---

> ### Author Response · Authors · 2023-11-21
> **Author Response (1/2)**
>
> Please see **Overall summary** at the top, which describes major changes, including dramatically improved performance on CIFAR-10 (from 91.7\% in the original manuscript to 92.7\% in this version), and a revamped introduction section that more clearly explains the motivation of our work.
>
> **Benefits of our approach**
>
> At present, there are three primary benefits of our approach.
>
> First, our work addresses a number of theoretical questions arising in the NNGP literature.  These include:
> * Is it possible to obtain an infinite-width, NNGP-like limit in which representation learning can occur?
> * Current NNGPs underperform relative to finite NNs.  One hypothesis that might explain this underperformance is that NNGPs have a fixed kernel, and thus lack representation learning.  While that hypothesis is plausible, it is not easy to confirm, given the huge differences between NNGPs and finite NNs.  To confirm this hypothesis, we really need a variant of the NNGP that somehow includes representation learning.  That's precisely what we develop in this paper.  And we show that by introducing representation learning into NNGPs, you indeed get a considerable performance boost (from 91.2\% in Adlam et al. (2023) to 92.7\% here). Furthermore, as seen in the updated results in Table 1, varying our regularisation strength allows us to smoothly interpolate between the NNGP regime and the representation-learning regime.
> All this suggests that it is indeed the lack of representation learning that is causing at least part of the underperformance of NNGPs relative to finite NNs. This has been hypothesised in the past (e.g. David MacKay. Introduction to gaussian processes. 1998) but we are the first to empirically show this on meaningful datasets like CIFAR-10.
>
> Second, our work forms a bridge, allowing us us to connect the fantastic collection of prior work that you point out in your review to current work on NNGPs.
>
> Third, improved performance against previous kernel methods.  Our best result is 92.7\% as opposed to previous results from kernel methods with parameters (89.8\%; Mairal 2016) and without parameters (91.2\%; Adlam et al. 2023).
>
>
>
> **Additional related work**
> Thanks for highlighting this awesome collection of related work!  We have updated the Related Work to include a thorough discussion of all of it (also see below).  We hope that our paper can drive renewed interest and attention towards this collection of prior work, by highlighting:
> * the potential excellent performance of deep kernel methods on datasets such as CIFAR-10
> * theoretical relationships between the broad family of deep kernel methods and "fashionable" topics, such as representation learning in infinite-width neural networks.
> In any case, we'd be excited to follow up with the authors of this work! (Anonymity permitting, of course.)
>
> At a high-level, there are two key differences to this past work.
>
> First, this work lacks a theoretical link to representation learning in infinite-width neural networks, and thus cannot be used to address the theoretical questions addressed by our work. But there are other differences in the details.
>
> Second, it seems that a lot of this work (e.g. Mairal et al. 2014, Mairal 2016, Johan 2017, Dexiong et al. 2019, Achten et al. 2023) uses a (convolutional/recurrent) neural network like structure to obtain a finite-dimensional projection/approximation of an underlying fixed kernel. In that sense, this work resembles (or could be used to get) an efficient approximation of NNGP-like kernels. In contrast, our objective (Eq. 5) is fundamentally not a projection/approximation of an underlying, fixed kernel.  Instead, it allows "full" flexibility in the Gram matrices/kernels, resembling exactly the type of flexibility in the kernel that you'd get in a deep GP or a finite, deep neural network.
>
> > [1] Bohn, B., Rieger, C., & Griebel, M. (2019). A representer theorem for deep kernel learning. The Journal of Machine Learning Research, 20(1), 2302-2333.
>
> Deep kernel learning is of course very different to our approach.
> In deep kernel learning, you train a neural network, and feed the outputs of that network into a kernel.
> In contrast, in a DKM approach, there is no neural network: there are no features and no weights.
> Instead, there are just kernels (or structured approximations to kernels).
>
> > Tonin, Francesco, et al. "Deep Kernel Principal Component Analysis for Multi-level Feature Learning." arXiv preprint arXiv:2302.11220 (2023),
>
> This is an unsupervised feature extraction method.

---

> ### Author Response · Authors · 2023-11-21
> **Author Response (2/2)**
>
> **Title too broad?**
> We agree that the term "deep kernel machine" is perhaps quite broad.  Your comment raises a bit of a dilemma: the use of the term "Deep Kernel Machine" in this context was introduced by Yang et al.  While their paper was formally published in 2023 at ICML, the paper (and terminology) was established on arXiv as early as 2021.  Moreover, there are other terms such as "Deep kernel learning" which seem very broad, but in practice are used to refer to a very specific way of combining a neural network and a kernel, ("Deep kernel learning" Wilson et al. 2015). Do you have any suggestions about how to resolve this dilemma?
>
> **Uncertainty**
> The three key advantages of our approach are outlined at the top of this response.
> Unfortunately, as we take the infinite-width limit, we lose the straightforward Bayesian interpretation, so the implied uncertainty properties are unclear, and we leave this for future work.
>
> **Minor aspects:**
>
>
> > Some typos and non-mathemcatical edits, e.g., the 2nd paragraph under Table 3 on page 8.
>
> Thanks, fixed!
>
> > In the provided codes, there are different variants of nonlinear kernels. But I didn't notice a clear descriptions on how these nonlinear units are chosen or considered.
>
> In practice, we just use the arccos kernel corresponding to the relu nonlinearity, to ensure a close correspondence to CNNs.
>
> > How about the empirical efficiency difference on varying the inducing points setups in Table 2?
>
> We have now added this to the paper in Table 2. We include the times below for the reviewers' convenience.
> | Number of Inducing Points          | Time per Epoch (seconds) |
> |------------------------------------|--------------------------|
> | 16/32/64                           | 45                       |
> | 32/64/128                          | 54                       |
> | 64/128/256                         | 82                       |
> | 128/256/512                        | 155                       |
> | 256/512/1024                       | 420                      |
> | 512/1024/2048                      | 1380                      |
>
>
> > In Table 1, many factors are investigated, and in different sections mostly the performances did not changge drastically in the compared techniques and were similar to other sections. The evaluation is comprehensive, but the conlusion is less focused, as we would hope to quickly get which factors are the most important ones. I these results don't change much and all attain good results, and then a mild suggestion or a global defaulting setting can be a take-away message.
>
> See the "Benchmark performance" paragraph for a description of how these experiments informed our choice of defaults for subsequent experiments.

---

> > ### Comment · Reviewer_hjfZ · 2023-11-22
> > **Thanks for the rebuttal**
> >
> > I would like to thank the authors for the replies. This work provides a link to the infinite-width neural network and uses some acceleration scheme via the usage of inducing points to make the kernel computation tractable. My concerns on the practical potential and the theoretical significance w.r.t. kernel tricks still remain, such as the empirical advantages (other benefits apart from the accuracy, e.g., hierarchical feature learning, more interpretability, etc can be considered, as simply the accuracy advantages can be reached by many methods) and the heritage of natural benefits from kernel trick or duality interpretation or Bayesian interpretation.
> >
> > On the scalability empirical experiments, I have no further requirement, as indeed such experimental setups are sufficient especially in the class of kernel methods, though some tuning parameters and the selection of  inducing points need to be determined and the model complexity and computation are worth further considerations.
> >
> > On the title, some key words related to Gaussian Process, infinite-widd neural network, Bayesian neural network can be involved, reflecting that this work is an extension upon this type of deep kernels in  (Yang et al. 2023).
> >
> > Up to the current version, I would like to keep my score; if in an acceptance, I would suggest the authors to further elaborate the discussions with the existing kernel methods, admitting their merits and also comparing with the focus in this work upon NNGP with infinite width, in the mean time articulate more also on the limitations of this work with outlooks for broader readers.

---

### Official Review · Reviewer_3QEZ · 2023-10-30

**Soundness:** 2 fair
**Presentation:** 1 poor
**Contribution:** 3 good
**Rating:** 5
**Confidence:** 3

**Summary:**

The paper extends the work of deep kernel machines with infinite-width Bayesian NN to convolutional networks. While the extension of the base model to convolutional networks is quite straightforward, making it efficient with inducing points is more challenging and is where the main technical novelty of this paper is.

**Strengths:**

- The work is a significant contribution to the field of infinite-width NN
- The extension to sparse convolutional DKM is novel
- The method has good results on several standard benchmarks.

**Weaknesses:**

- I found the definition of $C^\ell$ in Eq. 18 very confusing and unintuitive. This should be the inducing points equivalent of the BNN convolution operation in Eq. 8. However this involves summing over all of the inducing points in the previous layer. This is unusual and isn't part of any standard convolution.
- Experiments only show accuracy and NLL, and while this is fine for standard machine learning models, the benefit of the Bayesian approach is mostly in the other benefits it adds besides plain accuracy. It would make the paper much better if you show that your VI approximation still holds these properties with additional evaluations on these benchmarks like calibration or out-of-distribution detection.
- The writing needs improvement. English editing is needed to make the paper read like a more coherent text, instead of disjoint sentences.

**Questions:**

My main question is about Eq. 18 and the reason behind mixing all the previous inducing points.

---

> ### Author Response · Authors · 2023-11-21
> **Author Response**
>
> Please see **Overall summary** at the top, which describes major changes, including dramatically improved performance on CIFAR-10 (from 91.7\% in the original manuscript to 92.7\% in this version), and a revamped introduction section that more clearly explains the motivation of our work.
>
> **Eq. 18**
> Agreed: this isn't intuitive, but it is one of the key contributions in our framework.
> We could use inducing points that look like images/feature maps, but if we did that the memory usage would be far too high (as explained in the paper).
> Instead, we worked hard to come up with memory-efficient inter-domain inducing points that are just a flat vector, with no spatial structure.
> You can see that because the $F_{i,\lambda}^{\rm i}$ and $H^{\rm i}_{i', \mu}$ just have channel indices ($\lambda$ and $\mu$) or inducing point indices ($i$ or $i'$), and don't have any spatial indicies.
>
> But in Eq. 18, we have the convolutional weights, $W_{d \mu, \lambda}$, which does have spatial structure; how does that work?
> The answer is that the ``inner'' multiplication by C,
>
> $\sum_{i'} C_{di , i'} H^{\rm i}_{i', \mu}$
>
> has a result that has indices $i, d, \mu$.  i.e.\ this inner multiplication gives a bunch of image patches, which we can combine with the weights, $W_{d \mu, \lambda}$! In short, we multiply spatial-less inducing points by various weights C to form spatial patches, and then convolve these with the convolutional weights W.
>
>
> **Benefits of Convolutional DKMs**
> > Experiments only show accuracy and NLL, and while this is fine for standard machine learning models, the benefit of the Bayesian approach is mostly in the other benefits it adds besides plain accuracy. It would make the paper much better if you show that your VI approximation still holds these properties with additional evaluations on these benchmarks like calibration or out-of-distribution detection.
>
> The two key advantages of the DKM approach are described in the "Overall summary".  First, the DKM approach gives improved performance over previous kernel methods.  Our best result is 92.7\% as opposed to previous results from kernel methods with parameters (89.8\%; Mairal 2016) and without parameters (91.2\%; Adlam et al. 2023).  Second, our results have important implications for our theoretical understanding about the importance of representation learning in NNs (see modified Introduction). In particular, our new results show much more clearly in Table 1 how our regularisation strength hyperparameter allows the DKM framework to smoothly interpolate between the fixed NNGP kernel regime and the fully-expressive representation learning regime.
> Unfortunately, as we take the infinite-width limit, we lose the straightforward Bayesian interpretation, so the implied uncertainty properties are unclear, and we leave this for future work.
>
> **Writing**
> We have copy edited the paper with a native English speaker.

---

### Official Review · Reviewer_5yKS · 2023-11-01

**Soundness:** 3 good
**Presentation:** 2 fair
**Contribution:** 2 fair
**Rating:** 5
**Confidence:** 4

**Summary:**

This paper proposes an extension of deep kernel machines to leverage convolutions. It is based on the patch-based inter-domain inducing points in the convolutional GP literature. The main difference between convolutional DKMs and convolutional DGPs is that DKMs allow to perform representation learning--the Gram matrix is not fixed as in NNGPs but learnable, while regularized by a NNGP prior. Therefore, the model can have a large number of trainable parameters. Together with the convolutional structure, the model achieves 92% on CIFAR10, which outperforms previous convolutional GP methods and NNGP/NTKs.

**Strengths:**

* The proposed convolutional DKM model has strong performance on classification benchmarks. 92% accuracy on CIFAR10 is quite impressive if we classify DKM into the broad class of kernel methods.

* The technical development is sound--the proposed model formulation and training algorithms are sensible and computationally efficient.

**Weaknesses:**

* The contribution seems incremental. The proposed method is a combination of DKMs and the inter-domain inducing points used in convolutional (D)GPs.

* The objective for learning parametric gram matrices is regularized by a NNGP prior: $KL(N(0, G^\ell) \| N(0, K(G^{\ell-1}))$ computed from the gram matrix of the previous layer. Although this seems a sensible objective function to enable representation learning, it is unclear what the first principle is behind it. It is mentioned briefly in the paper "Formally, the DKM objective is the evidence lower bound (ELBO) for an infinite-width DGP in the Bayesian representation learning limit." It would be great to see a proof on this.

* The results are quite impressive if we categorize the proposed method into the "kernel method" class. However, I feel more evidences are needed to justify that this is a fair comparison. For example, the number of parameters in DKMs is much larger than NNGPs and convolutional (D)GPs due to the trainable gram matrices. It would be good to report these numbers in experiments and compare to a ResNet with the same number of parameters. Would the regularization bring any advantages over a such a ResNet?

* One big benefit of (D)GPs is that they can produce uncertainty estimates. Do convolutional DKMs exhibit similar capabilities? Are their predictions well-calibrated? Answering these questions could greatly strengthen the paper.

**Questions:**

Please see above questions.

---

> ### Author Response · Authors · 2023-11-21
> **Author response**
>
> Please see **Overall summary** at the top, which describes major changes, including dramatically improved performance on CIFAR-10 (from 91.7\% in the original manuscript to 92.7\% in this version), and a revamped introduction section that more clearly explains the motivation of our work.
>
> **Novelty of convolutional inter-domain inducing point method for DKMs**
> As discussed in the Introduction and Related Work, standard inducing point approaches for Convolutional GPs don't work, because they require features at each layer, while the DKM only gives us Gram matrices.  Working out how to get things to work efficiently, while working in the Gram matrix domain is highly non-trivial, and the key reason that the math in the paper is a bit painful.
>
> Additionally, the dramatic differences between the methods come out in the experiments.
> Our radically different approach from convolutional DGPs allows us to scale to much deeper models (e.g. Dutordoir et al. 2020 only went up to three layers), and hence allowed achieve dramatically improved performance (Dutordoir et al. only got 76.17\% on CIFAR-10, while we got 92.7\%).
>
> **Derivation of DKM objective**
> Please see Appendix A (which was present in the original manuscript) for a first-principles derivation, or the original DKM paper (Yang et al. 2023).
>
> **Appropriate baselines**
> You're right that the best comparison perhaps isn't the "parameter-free" NNGP methods.  The best comparison would be against other kernel methods that also have trainable parameters. We didn't originally include them, as there aren't many such methods, and they don't tend to perform as well as NNGPs.  Nonetheless, we've updated Table 3 to include such methods, including convolution DGPs (up to 76.17\% on CIFAR-10 with Dutordoir et al. 2020), and convolutional kernel networks (up to 89.8\% with Mairal, 2016). Our approach gives considerable performance improvements, with our best result being 92.7\%.
>
> **The benefits of convolutional DKMs**
> The two key advantages of the DKM approach are described in the "Overall summary".  First, the DKM approach gives improved performance over previous kernel methods.  Our best result is 92.7\% as opposed to previous results from kernel methods with parameters (89.8\%; Mairal 2016) and without parameters (91.2\%; Adlam et al. 2023).  Second, our results have important theoretical implications for our theoretical understanding about the importance of representation learning in NNs (see modified Introduction). In particular, our new results show much more clearly in Table 1 how our regularisation strength hyperparameter allows the DKM framework to smoothly interpolate between the fixed NNGP kernel regime and the fully-expressive representation learning regime, thus demonstrating that representation learning is a key component for successful deep models.
>
> Unfortunately, as we take the infinite-width limit, we lose the straightforward Bayesian interpretation, so the implied uncertainty properties are unclear, and we leave this for future work.

---

### Author Response · Authors · 2023-11-21
**Overall Summary**

## Summary

Our work resolves a key question at the heart of recent research on NNGPs.  Specifically, there is a large body of work in the NNGP and NTK literature, attempting to push up the performance of these methods.  There is good theoretical reason for doing this.  In particular, we want to know whether neural networks capture the "essence" of what makes neural networks work so well, or whether there is something missing.  Indeed, it does seem that NNGPs underperform relative to finite NNs, suggesting that NNGPs are indeed missing something fundamental. One hypothesis is that this missing piece is missing representation learning.  However, we can't really be sure that this is the case, given the numerous practical differences between NNGPs and finite NNs.  One way to confirm this hypothesis would be to develop an infinite-width NNGP-like approach that somehow includes representation learning.  We do exactly that, and find that we do indeed get improved performance over NNGPs (92.7\% vs the best NNGP result of 91.2\% from Adlam et al. 2023). What's more, the DKM framework allows us the smoothly interpolate between the NNGP regime and the fully-representation-learning regime by adjusting the regularisation strength parameter (see Table 1).

Moreover, our empirical result of 92.7\% gives a considerable improvement in the performance of previous kernel methods without parameters (91.2\% from Adlam et al. 2023) and with parameters (89.8\%, Mairal 2016).

Major changes from review:

We have substantially rewritten the introduction to emphasise the theoretical importance of our results to our understanding of representation learning in NNGP-like contexts.

The new 92.7\% result is a considerable improvement over the originally reported result of 91.7\%, obtained by running for an increased number of epochs (200). The 200 epoch runs on the largest model took 77 hours in total on a single A100 GPU. Note that this is still order(s) of magnitude faster than NNGP methods that need to compute all elements of a kernel matrix for all datapoints, and that we are working on methods to increase convergence speed (using preconditioning, for example) and decrease epoch time (using single precision floats) though this is of course out-of-scope for this first paper.

Perhaps more important than the highest accuracy, though, is that the effect of the regularisation strength in Table 1 is now much clearer, and smoothly makes the model less expressive as the regularisation strength (which encourages the model to be similar to the corresponding NNGP model) is increased. This clearly shows how the DKM framework allows one to smoothly interpolate between the inexpressive NNGP kernel regime ($\nu=\infty$) and the fully representation learning regime ($\nu \to 0$).

The reviewers were in agreement that the work was significant (3QEZ), novel (3QEZ,jw8S), technically sound (5yKS) and gave strong results (5yKS,3QEZ,hjfZ,jw8S):

5yKS:
* The proposed convolutional DKM model has strong performance on classification benchmarks.
* The technical development is sound--the proposed model formulation and training algorithms are sensible and computationally efficient.

3QEZ:
* The work is a significant contribution to the field of infinite-width NN
* The extension to sparse convolutional DKM is novel
* The method has good results on several standard benchmarks.

hjfZ:
* good results compared to other kernel techniques.
* the focus of this work is well presented.

jw8S:
* To my knowledge, the proposed inter-domain inducing point approach seems to be novel and work well in the given context.
* The authors conduct a comprehensive study of different architectures and hyperparameters of the proposed method.
* The paper is clearly written and well organized.

Given these unanimously positive comments, the disagreement in review scores is a bit surprising. We think the central issue is confusion around how the work is situated theoretically with respect to the NNGP literature. As such (and as discussed above), we've modified Introduction to emphasise that not only is the work novel, significant, technically sound and gives strong results, but the work also makes important theoretical contributions around our theoretical understanding of finite NNs and NNGPs. We hope this addresses the feedback from the reviewers, who otherwise seemed to recognise the value of our contributions.

---

### Meta-Review · Area_Chair_frjH · 2023-12-12

**Metareview:**

This paper develops a convolutional version of the recently introduced deep kernel machine (DKM). The convolutional DKM can scale up to CIFAR-sized datasets with reasonable efficiency and achieves better accuracy than previous kernels.

There have been efforts to push the performance limit of kernel methods due to their simplicity and theoretical properties. However, a gap remains between best-performing kernels and non-linear neural networks. This paper makes a solid contribution in this direction by developing a learnable kernel with good efficiency and SOTA accuracy on CIFAR-10/100. The reviewers had some concerns about the novelty and motivation of the work, but the area chair found that these questions were well addressed in the author rebuttal.

**Justification For Why Not Higher Score:**

Although there is some novelty, the method in the paper is based on a combination of previous techniques. Moreover, the accuracy improvement over previous (fixed) kernels is small.

**Justification For Why Not Lower Score:**

See meta-review.

---

### Decision · Program_Chairs · 2024-01-16

Accept (poster)